



Atmospheric
Chemistry
and Physics

# Characterising the seasonal and geographical variability in tropospheric ozone, stratospheric influence and recent changes

**Ryan S. Williams**[1], **Michaela I. Hegglin**[1], **Brian J. Kerridge**[2], **Patrick Jöckel**[3], **Barry G. Latter**[2], **and David A. Plummer**[4]

[1]University of Reading, Reading, UK
[2]Rutherford Appleton Laboratory (RAL), Harwell Campus, Didcot, UK
[3]Institut für Physik der Atmosphäre, Deutsches Zentrum für Luft- und Raumfahrt (DLR), Oberpfaffenhofen, Germany
[4]Canadian Centre for Climate Modelling and Analysis, Environment and Climate Change Canada, Montréal, QC, Canada

**Correspondence:** Ryan S. Williams (r.s.williams@pgr.reading.ac.uk)

**Abstract.** The stratospheric contribution to tropospheric ozone ($O_3$) has been a subject of much debate in recent decades but is known to have an important influence. Recent improvements in diagnostic and modelling tools provide new evidence that the stratosphere has a much larger influence than previously thought. This study aims to characterise the seasonal and geographical distribution of tropospheric ozone, its variability, and its changes and provide quantification of the stratospheric influence on these measures. To this end, we evaluate hindcast specified-dynamics chemistry–climate model (CCM) simulations from the European Centre for Medium-Range Weather Forecasts – Hamburg (ECHAM)/Modular Earth Submodel System (MESSy) Atmospheric Chemistry (EMAC) model and the Canadian Middle Atmosphere Model (CMAM), as contributed to the International Global Atmospheric Chemistry - Stratosphere-troposphere Processes And their Role in Climate (IGAC-SPARC) (IGAC–SPARC) Chemistry Climate Model Initiative (CCMI) activity, together with satellite observations from the Ozone Monitoring Instrument (OMI) and ozone-sonde profile measurements from the World Ozone and Ultraviolet Radiation Data Centre (WOUDC) over a period of concurrent data availability (2005–2010). An overall positive, seasonally dependent bias in 1000–450 hPa ($\sim 0$–5.5 km) sub-column ozone is found for EMAC, ranging from 2 to 8 Dobson units (DU), whereas CMAM is found to be in closer agreement with the observations, although with substantial seasonal and regional variation in the sign and magnitude of the bias ($\sim -4$ to $+4$ DU). Although the application of OMI averaging kernels (AKs) improves agreement with model estimates from both EMAC and CMAM as expected, comparisons with ozone-sondes indicate a positive ozone bias in the lower stratosphere in CMAM, together with a negative bias in the troposphere resulting from a likely underestimation of photochemical ozone production. This has ramifications for diagnosing the level of model–measurement agreement. Model variability is found to be more similar in magnitude to that implied from ozone-sondes in comparison with OMI, which has significantly larger variability. Noting the overall consistency of the CCMs, the influence of the model chemistry schemes and internal dynamics is discussed in relation to the inter-model differences found. In particular, it is inferred that CMAM simulates a faster and shallower Brewer–Dobson circulation (BDC) compared to both EMAC and observational estimates, which has implications for the distribution and magnitude of the downward flux of stratospheric ozone over the most recent climatological period (1980–2010). Nonetheless, it is shown that the stratospheric influence on tropospheric ozone is significant and is estimated to exceed 50 % in the wintertime extratropics, even in the lower troposphere. Finally, long-term changes in the CCM ozone tracers are calculated for different seasons. An overall statistically significant increase in tropospheric ozone is found across much of the world but particularly in the Northern Hemisphere and in the middle to upper troposphere, where the increase is on the order of 4–6 ppbv (5 %–10 %) between 1980–1989 and 2001–2010. Our model study implies that attribution from stratosphere–troposphere

exchange (STE) to such ozone changes ranges from 25 % to 30 % at the surface to as much as 50 %–80 % in the upper troposphere–lower stratosphere (UTLS) across some regions of the world, including western Eurasia, eastern North America, the South Pacific and the southern Indian Ocean. These findings highlight the importance of a well-resolved stratosphere in simulations of tropospheric ozone and its implications for the radiative forcing, air quality and oxidation capacity of the troposphere.

## 1 Introduction

Tropospheric ozone ($O_3$) has wide-ranging implications for air quality, radiative forcing and the oxidation capacity of the troposphere (Fiore et al., 2002a; Myhre et al., 2013). Whilst ozone is typically regarded as a pollutant at ground level, adversely affecting human health and ecosystems (Paoletti et al., 2014), it is a primary source of the hydroxyl (OH) radical which acts to cleanse the troposphere by breaking down a large number of pollutants, along with some greenhouse gases (Seinfeld and Pandis, 2006; Cooper et al., 2010). Despite this, ozone is also a greenhouse gas itself, exerting the largest radiative forcing in the upper troposphere due to the inherent low temperatures in the upper troposphere (Lacis et al., 1990). Since ozone has a relatively short global mean lifetime in the troposphere ($\sim$ 3 weeks), along with spatially and temporally highly varying sources and sinks (Lelieveld et al., 2009), it is not well mixed, with large spatial and temporal variations in ozone abundance as a result over seasonal, inter-annual and decadal timescales. This is reinforced by the strong dependence on sunlight as well as precursor emissions, which have both natural and anthropogenic sources (Cooper et al., 2014).

A large fraction of the ozone in the troposphere is formed through photochemical reactions of precursor molecules such as carbon monoxide (CO), nitrogen oxides ($NO_x$) and volatile organic compounds (VOCs). Since the late 19th century, however, changes in the tropospheric ozone burden can be largely attributed to anthropogenic precursor emissions, which have led to a significant increase in baseline (HTAP, 2010; Cooper et al., 2014) and also background (Fiore et al., 2002b; Zhang et al., 2008; Stevenson et al., 2013) ozone volume mixing ratios (VMRs), particularly in the Northern Hemisphere mid-latitudes (although it should be noted that this attribution is derived purely from modelling studies). Ozone may be produced either in situ or non-local to precursor source regions, as determined by the synoptic meteorology, with the potential for long-distance advection prior to photochemical destruction or deposition, given a lifetime of several weeks in the troposphere (Lelieveld et al., 2009). For instance, tropospheric ozone levels across western North America are particularly susceptible to increasing Asian emissions due to long-range transport across the Pacific (Hudman et al., 2004; Cooper et al., 2010; Lin et al., 2014, 2015). An additional influence is that of the exchange of stratospheric and tropospheric air masses, which leads to a net downward flux of ozone and a subsequent enhanced tropospheric ozone burden (Holton and Lelieveld, 1996; Lamarque et al., 1999), especially in mid-latitude regions (Miles et al., 2015).

Stratosphere–troposphere exchange (STE) of air is governed non-locally by the wave-driven large-scale meridional circulation, the Brewer–Dobson circulation (BDC) (Holton et al., 1995; Shepherd, 2007; Butchart, 2014). The BDC induces preferential troposphere-to-stratosphere transport (TST) in the tropics, in contrast to mid- to high-latitude regions where stratosphere-to-troposphere transport (STT) must prevail to conserve mass continuity (Holton et al., 1995). The BDC, and thus STE, exhibits strong seasonality in both hemispheres with the circulation strongest during wintertime but especially in the Northern Hemisphere, due to the largest wave-induced forcing occurring at this time (Holton et al., 1995). Given a photochemical lifetime of several months in the lower stratosphere, analogous to transport timescales, seasonality in the BDC results in a significant enrichment of ozone and other chemical tracers in the extratropical lower stratosphere over winter (Hegglin et al., 2006; Krebsbach et al., 2006); with the largest VMRs achieved close to the tropopause in early summer (Prather et al., 2011; Škerlak et al., 2014). Whilst it is recognised that the STE flux of ozone in the extratropics reaches a seasonal maximum in late spring and early summer (Yang et al., 2016), this incidentally coincides closely with the seasonal minimum in the downward STE mass flux of air (Škerlak et al., 2014; Yang et al., 2016). This strongly implies that the ozone VMR at the tropopause controls the seasonality in the downward ozone flux. Staley (1962) was the first to note that it is in fact the displacement of the tropopause altitude seasonally in each hemisphere that primarily governs the downward mass flux: maximum in spring as the tropopause rises and minimum in autumn as the tropopause falls relative to the average state. Analysis of deep STE events, where direct entrainment of stratospheric air into the planetary boundary layer (PBL) occurs, indicates that the downward transport of ozone is primarily controlled by the mass flux for these events, with a peak in early spring (Škerlak et al., 2014).

Whilst it is accepted that STE is an important and significant source of upper-tropospheric ozone (e.g. Holton et al., 1995), the influence on near-surface ozone levels is poorly understood. Globally, Lamarque et al. (1999) estimated that STE increases the average tropospheric column amount by only a modest $\sim$ 11.5 % using a three-dimensional global chemistry transport model. However, on a monthly resolved basis, this influence was shown to increase to $\sim$ 10 %–20 % in the lower troposphere and $\sim$ 40 %–50 % in the upper troposphere. More recent modelling studies, however, show a much larger influence. The annual mean estimated influence of the stratosphere is shown to range between 25 %

and 50 % in the lower and middle extratropical troposphere, with the largest influence in the Southern Hemisphere where other sources of ozone provide a smaller contribution to the tropospheric ozone budget, according to various modelling studies (e.g. Lelieveld and Dentener, 2000; Banerjee et al., 2016). Hess and Zbinden (2013) found from observations that lower-stratospheric (150 hPa) ozone explains nearly 70 % of the variance in mid-troposphere (500 hPa) ozone trends and variability over Northern Hemisphere mid-latitude regions, including Canada, the eastern US and Northern Europe. Furthermore, a number of mid-latitude case studies have demonstrated that STT events may provide a much larger contribution to surface ozone in some seasons (typically spring) and more locally on timescales of hours to days given favourable meteorological conditions. Over a 3-month period between April and June 2010, Lin et al. (2012) concluded that the stratosphere was the source of 20 %–30 % of surface $O_3$ across the western US using the high-resolution ($\sim 50 \times 50$ km$^2$) GFDL AM3 chemistry–climate model (CCM), with episodic enhancements of some 20–40 ppbv of the surface maximum 8 h average (MD8A) ozone estimated from 13 identified stratospheric intrusion events. Similarly, model-based studies find evidence for a significant stratospheric contribution to the pronounced tropospheric summertime ozone maximum over the eastern Mediterranean and the Middle East (EMME) (Zanis et al., 2014; Akritidis et al., 2016) and the Persian Gulf (Lelieveld et al., 2009), with influence as far down as the PBL where near-surface ozone levels are known to frequently exceed EU air quality standards.

Observationally based studies show a wide range in the level of stratospheric influence. In conjunction with a beryllium (Be)-based mixing model, Dibb et al. (1994) showed that the stratosphere has a maximum influence during spring in the Canadian Arctic of a mere 10 %–15 % at the surface. Greenslade et al. (2017) also found only a small stratospheric contribution (1 %–3.5 %) to the mean tropospheric ozone burden for three sites neighbouring the Southern Ocean, although with exceedances of 10 % during individual events. A number of Europe-focussed studies highlight the significance of the stratosphere during episodic events, particularly over Alpine regions where elevated regions are sometimes directly impacted by stratospheric intrusions (e.g. Stohl et al., 2000; Zanis et al., 2003; Colette and Ancellet, 2005). This influence is typically largest in winter and spring (smallest in summer), although the seasonality exhibits greater complexity at some high-altitude locations, which is largely site-dependent. Significant enhancements in surface ozone, in association with stratospheric intrusions, have also been detected across the Himalayas during winter especially (up to 25 % contribution), in direct contrast to minimal influence during the summer monsoon season (e.g. Cristofanelli et al., 2010). Summertime ozone-sonde campaign measurements over the north-eastern US (Thompson et al., 2007a, b) imply a stratospheric contribution of $\sim 20$ % to 25 % to the

tropospheric column ozone during summer 2004, which is comparable to the budget inferred from European profiles (Colette and Ancellet, 2005). A similar level of influence is found on average in the middle and upper troposphere for 18 North American sites based on summer ozone-sonde campaign data between 2006 and 2011 (Tarasick et al., 2019). Ozone-sonde measurements from all seasons between 2005 and 2007 reveal a larger influence still (34 % or 22 ppbv) over south-eastern Canada, decreasing to 13 % (5.4 ppbv) and 3.1 % (1.2 ppbv) in the lower troposphere and boundary layer respectively, with typical occurrence of STT of 2–3 days (4–5 days) during spring and summer (autumn and winter).

The current understanding of the seasonal and regional climatology of tropospheric ozone is severely constrained by the paucity of in situ measurements from ozone-sondes and aircraft measurements, which are spatially and temporally biased, although the advent of satellite remote-sensing platforms in recent years for the inference of global tropospheric ozone abundance has reduced uncertainty to a significant extent (Parrish et al., 2014). Relatively long ($\sim$ decadal) global satellite datasets of tropospheric ozone now exist from several platforms (e.g. the Ozone Monitoring Instrument, OMI; the Tropospheric Emission Spectrometer, TES; the Total Ozone Mapping Spectrometer, TOMS; the Microwave Limb Sounder, MLS) TS1 that have been extensively validated with respect to in situ and ground-based remote-sensing measurements as well as inter-satellite comparisons. Nonetheless, there are inherent limitations with retrieving tropospheric ozone from spaceborne instruments, and this has implications for the accuracy of resultant satellite-based climatologies (Gaudel et al., 2018). Scientists, however, require tools such as CCMs, which offer sensitivity simulations and specific diagnostic variables that are not available from observations alone, to elucidate the drivers of variability and longer-term changes in the global distribution of tropospheric ozone, which includes the quantification of the stratospheric influence. Additionally, CCMs can be used to assess and quantify the causes of tropospheric ozone features through the analysis of photochemical production and loss rates, together with transport tracer simulations. The latter can serve to identify the relative importance of in situ photochemical production, long-range transport and stratospheric influence. Nonetheless, such simulations are subject to a number of constraints, including limitations in model horizontal and vertical resolution, complexity of the implemented chemistry scheme, and the realism of simulated transport characteristics. Above all, however, the largest unknown by far is the accuracy of the precursor emission inventories used in CCM simulations (Hoesly et al., 2018).

In this study, the seasonal climatology, inter-annual variability and long-term evolution of the influence of stratospheric ozone on tropospheric ozone and its geographical dependencies is investigated with the aim to update and extend the findings of Lamarque et al. (1999). A summary of the dif-

ferent data sources used is given in Sect. 2. As a first step in Sect. 3, we test the realism of two state-of-the-art CCMs by comparing their ozone estimates with the ozone distributions derived from OMI satellite measurements over a common baseline period, together with spatially and temporally limited vertical profile information provided by ozone-sondes. Noting the model biases with respect to the observations, the fine-scale vertical resolution offered by the CCMs is then exploited to analyse regional and seasonal variations in the vertical distribution of $O_3$ in Sect. 4, together with ozone of stratospheric origin ($O_3S$) and the relative contribution of $O_3S$ to the total amount of $O_3$ (the stratospheric ozone fraction: $O_3F$) to infer the importance of the stratosphere in determining tropospheric ozone levels. Finally, height-resolved seasonal changes in model $O_3$ and $O_3S$ are examined globally between 1980–1989 and 2001–2010 in Sect. 5. The findings presented in both Sects. 3 and 4 are discussed within the context of the wider literature. Finally, Sect. 6 will provide a summary of the findings, along with an overview of the utility of the models for improving our understanding of the spatial distribution and changes in tropospheric ozone.

## 2 Data sources

### 2.1 CCM simulations

This study uses RefC1SD CE1 specified-dynamics hindcast simulations, conducted for the Chemistry Climate Model Initiative (CCMI-1) (Hegglin and Lamarque, 2015; Morgenstern et al., 2017), of both ozone ($O_3$) and stratosphere-tagged tracer ozone ($O_3S$) for the period 1980–2010 inclusive from two state-of-the-art CCMs: European Centre for Medium-Range Weather Forecasts – Hamburg (ECHAM)/Modular Earth Submodel System (MESSy) Atmospheric Chemistry (EMAC; Jöckel et al., 2016) and the Canadian Middle Atmosphere Model (CMAM; Scinocca et al., 2008). These two models were primarily selected due to the close similarity in the $O_3S$ tracer definition (detailed below in Sect. 2.1.1 and 2.1.2 respectively), which is either absent or defined differently in other CCMI models and is fundamental to the quantification of the stratospheric influence and attribution to recent changes in tropospheric ozone in this study. $O_3S$ decays according to the same reactions used in the $O_3$ simulations, although the reactions leading to photochemical production of ozone are omitted for the $O_3S$ tracers (Roelofs and Lelieveld, 1997). In each simulation, the prognostic variables: temperature, vorticity and divergence as well as (the logarithm of) surface pressure for ECHAM only (the coupled general circulation model in EMAC) from the ERA-Interim reanalysis dataset have been used to nudge each CCM towards the observed atmospheric state through Newtonian relaxation, with corresponding relaxation times of 24, 6, 48 and 24 h respectively for EMAC (Jöckel et al., 2016) and 24 h for all three variables in CMAM (McLan-

dress et al., 2013). Variability in sea surface temperatures (SSTs) and sea ice concentration is directly accounted for in both EMAC and CMAM from ERA-interim and HadISST (provided by the UK Met Office Hadley Centre) respectively (Rayner et al., 2003; Morgenstern et al., 2017). Furthermore, each model includes either prescribed decadal emissions or lower boundary conditions of anthropogenic and natural greenhouse gas (GHG) and ozone precursor emissions (which act as a forcing) from the MACCity inventory, which is based on the Coupled Model Intercomparison Phase 5 (CMIP5) inventory and representative concentration pathway (RCP) projections (Lamarque et al., 2010; Hoesly et al., 2018), alongside variability induced by other natural forcings such as solar activity and volcanic eruptions in most simulations (Brinkop et al., 2016). All simulations used are compliant with the CCMI definitions specified by the International Global Atmospheric Chemistry (IGAC) and Stratosphere-troposphere Processes And their Role in Climate (SPARC) communities (Eyring et al., 2013). The stratospheric influx for CCMI models ranges from $\sim 400$ to $650\,\mathrm{Tg\,yr^{-1}}$, which is within the range estimated from observational studies (IPCC, 2013). For full details of the model chemistry treatments and emission inventories used, the reader is directed to the CCMI review paper by Morgenstern et al. (2017) as well as Jöckel et al. (2016) for EMAC and the relevant section of Pendlebury et al. (2015) for CMAM. The main difference between the two models is the complexity of the tropospheric chemistry scheme, namely that CMAM simulates no non-methane hydrocarbon chemistry, with additional differences in the model transport schemes, the treatment of heterogeneous chemistry, and accounting for $NO_x$ and isoprene emissions and representation of the Quasi-Biennial Oscillation (QBO). A brief overview of the two models and these differences is provided below (Sect. 2.1.1 and 2.1.2).

### 2.1.1 EMAC

RC1SD-base-10 simulation results (without nudging of the global mean temperature) from the interactively coupled EMAC model are used in this study, which have a T42 (triangular) spectral resolution, equating to a quadratic Gaussian grid of $\sim 2.8°$ by $2.8°$ and 90 vertical hybrid sigma pressure levels up to 0.01 hPa (Jöckel et al., 2016). EMAC uses the flux-form semi-Lagrangian (FFSL) transport scheme for chemical constituents, water vapour, cloud liquid water and cloud ice (Lin and Rood, 1996), with the chemistry submodels MECCA (Sander et al., 2011) and SCAV (Tost et al., 2006) describing the kinetic systems in the gaseous and aqueous or ice phase respectively. Comprehensive atmospheric reaction mechanisms that include basic $O_3$, $CH_4$, $HO_x$ and $NO_x$ chemistry, non-methane hydrocarbon (NMHC) chemistry up to $C_4$ and isoprene, halogen (Cl and Br) chemistry, and sulfur chemistry is all included in the chemical scheme. Relevant for the representation of heterogeneous chemistry in the stratosphere, deviations from thermodynamic equilib-

rium are accounted for, which has implications for the distribution of polar stratospheric clouds (PSCs) and associated ozone depletion. In the troposphere, an offline representation of aerosol (dust, sea salt, organic carbon, black carbon, sulfates and nitrates) provides surfaces for heterogeneous chemistry. Emissions of lightning $NO_x$, soil $NO_x$ and isoprene ($C_5H_8$) are parameterised online for EMAC using the submodel ONEMIS (Kerkweg et al., 2006; Jöckel et al., 2016). The model provides a consistent handling of the photolysis (submodel JVAL; Sander et al., 2014) and shortwave radiation schemes (submodel FUBRAD; Kunze et al., 2014), with particular regard to the evolution of the 11-year solar cycle (Morgenstern et al., 2017). The QBO is internally generated by the model, although zonal winds near the Equator are nudged towards a zonal wind field (Brinkop et al., 2016) with a 58-day relaxation timescale to ensure realistic simulation of the QBO magnitude and phasing (Jöckel et al., 2016). For tracing stratospheric ozone, an additional diagnostic tracer $O_3S$ is reset to the standard ozone tracer above the tropopause (using the World Meteorological Organization, WMO, thermal definition equatorward of 30° N and S and using the 3.5 potential vorticity unit, PVU, dynamical tropopause definition poleward of 30° N and S) as defined in every model time step. The $O_3S$ tracer is transported across the tropopause and subject to the tropospheric ozone sink reactions. The corresponding chemical loss of $O_3S$ ($LO_3S$) is diagnosed and integrated and, in addition to its dry-deposition, provides a direct measure for the stratosphere-to-troposphere exchange of ozone (Jöckel et al., 2006, 2016).

### 2.1.2 CMAM

Simulations from the atmosphere-only CMAM are used here with a T47 spectral resolution (equivalent to $\sim 3.75°$ by 3.75°) on the linear Gaussian grid used for the physical parameterisations in CMAM, with 71 vertical hybrid sigma pressure levels which extend to 0.01 hPa (Hegglin et al., 2014; Pendlebury et al., 2015). The model uses spectral advection of "hybrid" moisture for transport (Merryfield et al., 2003) and a similar spectral advection of "hybridized" tracers for chemically active tracers exhibiting strong horizontal gradients (Scinocca et al., 2008). Whilst a representation of heterogeneous chemistry on PSCs is provided, the model does not account for nitric acid trihydrate (NAT) or polar stratospheric cloud (PSC) sedimentation (resulting in denitrification). Heterogeneous chemistry calculations are also made in the troposphere through prescribing sulfate aerosol surface area densities. Chemistry is calculated throughout the troposphere, although the only hydrocarbon considered is methane. To account for isoprene ($C_5H_8$) oxidation in CMAM, an additional 250 Tg-CO yr$^{-1}$ in emissions (including an additional 160 Tg-CO yr$^{-1}$ from soils) is included, distributed as Guenther et al. (1995) isoprene emissions. Unlike EMAC, soil $NO_x$ emissions are not calculated online for CMAM and are instead prescribed, with lightning $NO_x$ emis-

sions parameterised from the Allen and Pickering (2002) updraft mass flux scheme (Morgenstern et al., 2017). In contrast to EMAC, consistency in the radiation and photolysis schemes has not specifically been imposed. Although CMAM does not generate a QBO internally, a representation of the QBO is induced in the specified-dynamics simulations through nudging to ERA-Interim. The stratospheric ozone ($O_3S$) tracer uses the WMO thermal tropopause definition as the threshold for tagging ozone as stratospheric across all latitudes, with an additional criterion that the tropopause must be $< 0.7$ in hybrid-sigma coordinates to prevent erroneous identification at high latitudes, during winter especially. Every time step, the $O_3S$ tracer is set equal to the model ozone above the tropopause, while below the tropopause the $O_3S$ tracer has an imposed first-order loss rate equal to the model-calculated first-order chemical loss rate of $O_x$ defined as $O_x = O_3 + O(^1D) + O(^3P) + NO_2 + HNO_4 + 2 \times NO_3 + 3 \times N_2O_5$. The $O_3S$ tracer also undergoes dry deposition at the surface with the same dry-deposition velocity as calculated for ozone.

## 2.2 Observations

### 2.2.1 OMI

OMI is a Dutch–Finnish UV–visible nadir-viewing solar backscatter spectrometer aboard the NASA-Aura satellite launched in July 2004. The satellite has a retrograde, sun-synchronous polar orbit (inclination of 98.2°) at an altitude of 705 km, providing some 14 orbits a day with a local equatorial crossing time in the ascending node of 13:45 local time (Levelt et al., 2006). OMI operates in the 270–500 nm spectral interval TS2 and has a spectral resolution of 0.42–0.63 nm (Foret et al., 2014). OMI is the first of a generation of instruments which use 2-D detector arrays, providing concurrent sampling at all across-track positions, as opposed to platforms which use a 1-D detector array to scan across track. OMI supplements the observational knowledge of ozone from other longstanding satellite platforms, such as NASA's TOMS instrument and ESA's Global Ozone Monitoring Experiment (GOME) instrument, at a much enhanced spatial resolution (e.g. 13 km × 24 km for OMI compared with 40 km × 320 km for GOME in the along-track and across-track directions nominally at nadir). The across track resolution, however, becomes significantly coarser away from nadir; reaching 13 km × 150 km towards the edge of the swath (corresponding to an angle of 57° from nadir). The swath is 2600 km wide at the surface resulting from a wide field of view of 114°, with a near-global coverage time of 1 day (Levelt et al., 2006; Foret et al., 2014). Temperature-dependent spectral structure in the region between 320 and 345 nm (the Huggins Band) contains the information required for the retrieval of ozone in the troposphere region (Miles et al., 2015). The logarithm of the ozone (VMR) on a fixed pressure grid (surface pressure, 450, 170,

100, 50, 30, 20, 10, 5, 3, 2, 1, 0.5, 0.3, 0.17, 0.1, 0.05, 0.03, 0.017, 0.01 hPa) provides the basis for the retrieved profiles (Miles et al., 2015).

This study uses 1000–450 hPa (0–5.5 km) sub-column ozone values retrieved from OMI, as derived using the Rutherford Appleton Laboratory (RAL) height-resolved optimal estimation profiling scheme (Miles et al., 2015; Gaudel et al., 2018) for one in four $50 \times 50$ km samples in every $100 \times 100$ km bin, which has been further optimised to increase sensitivity to tropospheric ozone. These "Level-2" (L2) data have been averaged into monthly mean $2.5° \times 2.5°$ ($\sim 275$ km) gridded "Level-3" (L3) data between 2005 and 2010. This resolution is more comparable with the resolution of the CCM simulations used in this study for model comparisons (Sect. 3). Validation against ozone-sondes for this sub-column, after applying averaging kernels (AKs) to account for vertical smearing associated with the satellite retrieval, yields a relatively low retrieval bias of $\sim 1.5$ Dobson units (DU) (6 %) (Miles et al., 2015). The sign of the bias is latitude-dependent for lower-tropospheric ozone: there is underestimation in the Southern Hemisphere by $\sim 15$ %–20 % (1–3 DU) and overestimation in the Northern Hemisphere by $\sim 10$ % (2 DU). These systematic biases can be attributed to inaccuracies in the radiative transfer modelling, which are partially rectified through the use of a priori information to shift the erroneous retrieved profiles towards the true values (Mielonen et al., 2015). An additional monthly mean, (linearly interpolated) latitude-dependent bias, identified with respect to the global ozone-sonde ensemble, was also corrected for in the OMI data used in this study. Other filtering criteria used to enhance the quality of the dataset include omission of observations with a cloud fraction greater than 0.2 and a solar zenith angle exceeding 80°. This estimation differs from other techniques such as cloud slicing (e.g. Ziemke and Chandra, 2012) and residual methods such as total-column ozone (TCO) from OMI minus vertical profile measurements from the MLS TS3 (e.g. Ziemke et al., 2011). In comparison with the OMI–MLS method, the OMI–RAL profiling scheme is more (less) sensitive to the lower (upper) troposphere (Gaudel et al., 2018). To ensure a direct comparison with other datasets in order to test the level of agreement with models and ozone-sonde observations, AKs should be applied to induce such smearing of information that inherently occurs in UV–nadir satellite measurements. The influence of AKs is critically evaluated for the 1000–450 hPa sub-column for both the models and ozone-sondes in Sect. 3.

OMI is regarded as a very stable platform, with the radiometric degradation during the instrument's lifetime estimated to have been just $\sim 2$ % in the UV and $\sim 0.5$ % in the visible channel, which is significantly lower than other comparable instruments (Levelt et al., 2018). Despite this, the quality of radiance data began to decline from 2007 onwards (but particularly starting from 2009) across all wavelengths in a progressively larger number of across-track views, corresponding to rows in the 2-D detector arrays, suspected to be blocked by insulation blankets covering the instruments which have become damaged. This one main anomaly is subsequently referred to as the row anomaly (Schenkeveld et al., 2017). Although OMI has relatively high sensitivity to the troposphere, sensitivity is much weaker near the surface due to the limited penetration of photons and subsequent reduced signal in the backscattered radiance spectrum (Sellitto et al., 2011), with factors such as surface albedo and aerosols in the PBL also resulting in additional interference (Liu et al., 2010).

### 2.2.2 Ozone-sondes

Vertical ozone profile data over the period 1980–2010 were derived from the World Ozone and Ultraviolet Radiation Data Centre (WOUDC); an archive of balloon-borne in situ measurements of ozone, together with other variables such as temperature, humidity and pressure. Ozone-sondes typically provide a vertical resolution of $\sim 150$ m from the surface up to a maximum altitude of approximately 35 km, although not in all cases (Worden et al., 2007; Nassar et al., 2008). Most sonde stations launch ozone-sondes on a weekly basis, but a number of European sites provide measurements 2–3 times a week (Worden et al., 2007). The WOUDC archive contains measurements from primarily electrochemical concentration cell (ECC) sondes, but also from two other instruments: the Brewer–Mast (BM) and the Japanese ozone-sonde (KC) (SPARC, 1998), which all yield measurements of ozone equivalently. The reader is directed to Liu et al. (2013) for further details of the WOUDC measurement network, including a map and table of all observation sites. The accuracy of sonde measurements is typically estimated to be within the range of ±5 %, depending on various factors. Precision between the various sonde types is estimated to be within ±3 %, with systematic biases of less than ±5 % within the lower to middle stratosphere (12–27 km altitude range), provided that profile measurements have been normalised with respect to ground-based total ozone measurements (SPARC, 1998).

Uncertainties are, however, much larger in the troposphere due to lower ozone VMRs, yielding a relatively low signal-to-noise ratio, which increases the susceptibility to both instrumental errors and instrumental variability. Sonde performance can additionally be affected by local air pollution, which can further enhance the level of uncertainty. Systematic differences between different instruments in the troposphere were estimated to vary between 10 % and 15 % in various intercomparison campaigns between 1970 and 1990 (Beekman et al., 1994; Smit et al., 1998). There is evidence that the ECC sondes have greater precision and consistency than either the BM or KC sondes here (e.g. WMO-III and JOSIE campaigns), i.e. a precision of ±5 %–10 % for ECC compared with a range of 10 %–20 % for BM and KC. A small positive bias of 3 % is noted for ECC with no evidence of biases exceeding ±5 % for BM and KC (Smit and Straeter, 2004a, b).

## 3 Tropospheric ozone (model–measurement comparison)

In order to evaluate the utility of the models in assessing tropospheric ozone and estimating stratospheric influence, the CCM simulations (EMAC and CMAM) are first validated here against the OMI observations, in addition to the spatially and temporally limited, height-resolved ozone-sonde measurements. This is achieved through a combined model–measurement characterisation of the seasonal and geographical variability of tropospheric ozone (Sect. 3.1), together with the inter-annual variability (Sect. 3.2) over the 2005–2010 period. Lastly, a vertically resolved assessment of the CCMs is provided for three different mid-latitude regions (Europe, eastern North America and the Tasman Sea) from aggregated ozone-sonde profile measurements between 1980 and 2010 (Sect. 3.3).

Seasonal composites of monthly mean 1000–450 hPa (0–5.5 km) sub-column ozone from OMI, together with available ozone-sonde-derived AK-fitted sub-columns, and the respective differences for each AK-fitted CCM are shown in Fig. 1. A seasonal maximum in tropospheric ozone is evident in each hemisphere during spring, which is more pronounced in the Northern Hemisphere and extended in many regions through to summer (JJA). In contrast to the extratropics, tropospheric ozone remains low year-round ($< 20$ DU) at low latitudes although some seasonality is apparent, notably a northward shift in the region of lowest ozone from boreal winter into summer and the reverse from boreal summer back to winter. This is likely associated with the seasonal migration of the Inter Tropical Convergence Zone (ITCZ), which closely follows the region of maximum solar insolation. In this region, strong upwelling occurs which leads to the transport of ozone-depleted air from the tropical PBL upwards towards the tropopause. This is most pronounced across the Maritime Continent where convective activity is climatologically most intense (e.g. Thompson et al., 2012).

The BDC, which leads to meridional transport in ozone and other constituents in the stratosphere, is strongest during winter (weakest during summer), and it is this annual variability which exerts a major influence over the seasonality of free-tropospheric ozone (through changes in STE), in regions of the extratropics where emissions of tropospheric ozone precursors are at a relatively low background level (Roscoe, 2006). This is invariably the case across much of the Southern Hemisphere, where anthropogenic precursor emissions are substantially lower and more spatially confined in comparison with the Northern Hemisphere. In some regions such as the South Atlantic, it is evident that tropospheric ozone is similarly high in winter (JJA) ($\sim 25$–30 DU), but it is known that this is a result of biomass burning activity in western Africa and resultant plumes, which are advected offshore during the dry season in particular (e.g. Mauzerall et al., 1998). Across Antarctica and the Southern Ocean, however, halogen-induced stratospheric ozone depletion is likely the dominant driver of the seasonality, leading to a minimum in spring (SON), although no observations from OMI are available during the polar night (MAM and JJA). In the Northern Hemisphere, the strong influence of emission precursors from widespread anthropogenic activity serves to delay and broaden the maximum, since the peak in the in situ photochemical formation of ozone is driven by solar insolation. This is particularly apparent in subtropical regions such as the eastern Mediterranean, due to favourable photochemical conditions for the production and subsistence of ozone during the summer months.

A corresponding zonally averaged monthly mean evolution, together with the respective differences for each CCM (both with and without AKs), is additionally shown in Fig. 2 and further summarised as 30° latitude band averages in Table S1 in the Supplement. Whilst the AK-fitted EMAC differences with respect to OMI (Figs. 1 and 2d) show an overall year-round, albeit seasonally varying positive bias, particularly within the 0 to 30° latitude band ($\sim +2$–8 DU), the difference is largely negative in CMAM ($\sim 0$ to $-4$ DU), except during spring (MAM) in parts of the Northern Hemisphere ($\sim 0$ to $+4$ DU) and within the 30 to 60° S latitude band ($\sim +2$–6 DU). Although such differences on a zonally averaged basis are relatively small (on the order of 10 %–20 %), the systematic nature and seasonal dependence of such biases is important to consider. Regional differences are evidently larger, however, with differences of up to 10 DU (50 %), such as over mid-latitude oceanic regions where both CCMs show a positive bias relative to OMI and also with respect to limited available ozone-sonde data from maritime locations. Some continental regions such as eastern Asia on the other hand show a negative bias in most seasons, with the largest in winter (DJF) (5–10 DU or 20 %–40 %). A recent study by Hoesly et al. (2018) shows discrepancies between the CMIP5 $NO_x$ emissions database (used in CCMI emission inventories) and an updated, refined database over the time frame considered, the Community Emissions Data System (CEDS), which could explain the pattern of biases between the continental regions of the Northern Hemisphere. Whilst the CMIP5 emissions dataset is composed of "best available estimates" from many different sources, the dataset has limited temporal resolution (10-year intervals), contains inconsistent methods across emission species, and lacks uncertainty estimates and reproducibility. The CEDS dataset addresses some of these shortcomings by also factoring in activity data to estimate country-, sector- and fuel-specific emissions on an annual basis, which is further calibrated to existing inventories through emission factor scaling. The sign of the biases is more complex and spatially variable in summer (JJA) but are typically low ($-3$ to $+3$ DU), implying that the CCMs are reasonably consistent overall with the OMI measurements during this season. In the Southern Hemisphere, the general positive bias is weaker (particularly in austral winter and spring), and most regions show a negative bias in at least one season. Model–measurement agreement

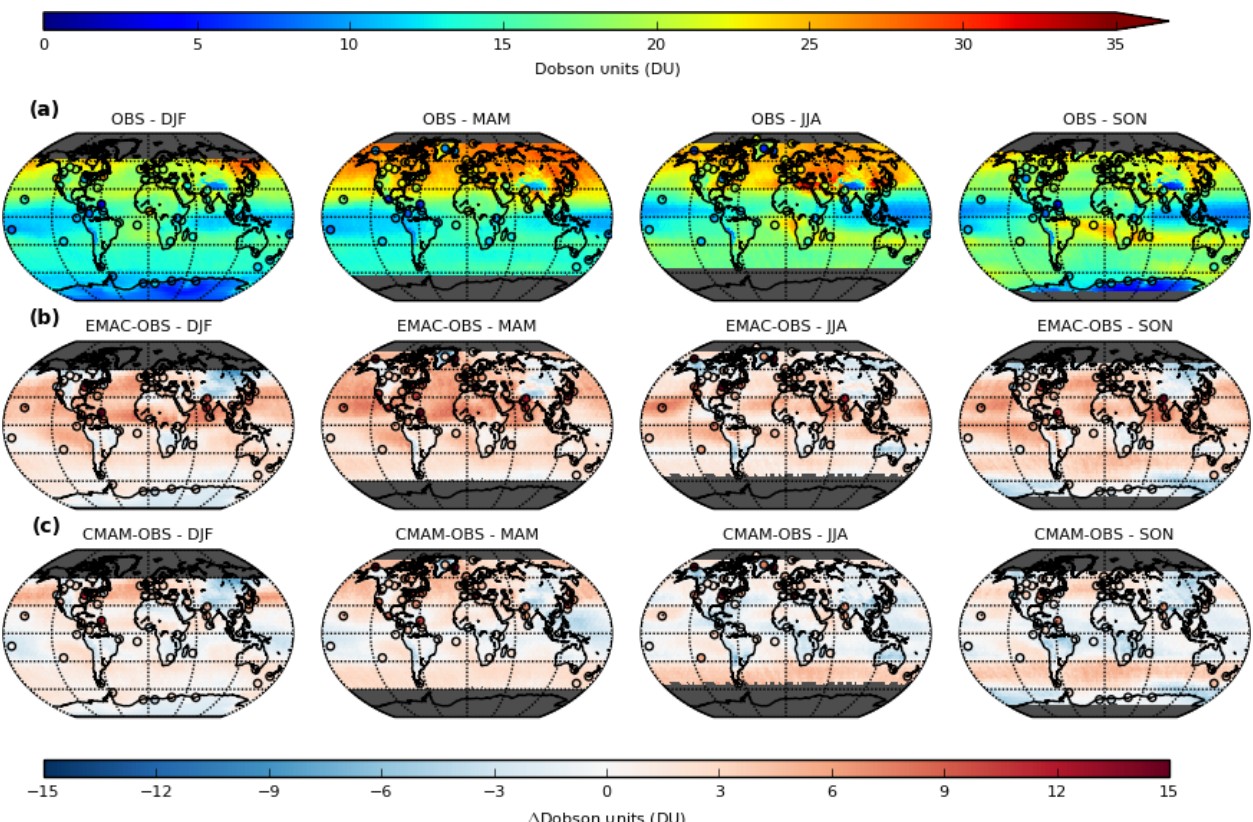

**Figure 1.** Seasonal composites of monthly averaged 1000–450 hPa (0–5.5 km) sub-column $O_3$ (DU) (left to right) for 2005–2010 from **(a)** OMI, **(b)** EMAC minus OMI and **(c)** CMAM minus OMI. Circles denote **(a)** equivalent ozone-sonde-derived sub-column $O_3$ (DU), **(b)** EMAC minus ozone-sonde differences and **(c)** CMAM minus ozone-sonde differences. All data were regridded to 2.5° resolution ($\sim$ 275 km). All model and ozone-sonde sub-column data have been modified using AKs to ensure a direct comparison.

here is typically higher compared with the Northern Hemisphere, particularly for latitudes where $O_3$ precursor emissions are lower and in the less photochemically active seasons (i.e. autumn and winter). This could indicate that CCMs simulate excessive photochemical production of ozone in the Northern Hemisphere particularly (Young et al., 2013; Shepherd et al., 2014) or that the role of tropospheric sinks (e.g. through wet and dry deposition or other loss reactions) is underestimated (Revell et al., 2018), with our results indicating regionally differing magnitudes in these biases.

Both Fig. 2 and Table S1 show the importance of applying AKs (on a monthly mean zonally averaged basis) in order to diagnose the agreement between the two datasets, by enabling a like-for-like comparison, since it is clear that both CCMs significantly underestimate the amount of tropospheric ozone overall at both middle and high latitudes, relative to the OMI observations (Fig. 2b–c). The effect of applying the AKs (Fig. 2d–e) is shown to significantly reduce or even eradicate the negative bias (poleward of 30° N and S), and it is this difference which indicates the approximate magnitude of the influence vertical smearing has on the retrieved OMI sub-column measurements. A residual negative bias ($\sim -2$ to $-6$ DU) also exists in the Southern Hemisphere during spring (SON) over the Southern Ocean south of 60° S (adjacent to Antarctica). This might relate to differences in the representation of a transport barrier such as the edge of the wintertime polar vortex, which influences mixing in the surf zone region and is eradicated in this season, together with disparities in the magnitude of the Antarctic ozone hole, which has implications for vertical smearing, influencing the resultant tropospheric ozone burden. Indeed, a cold-pole bias which leads to a delayed onset in the seasonal breakdown of the polar vortex is an inherent bias common to most CCMs (McLandress et al., 2012). Biases in much of the tropics appear also to be connected to dynamics which favour long-range transport (e.g. trade wind circulations) originating from regions of known precursor emissions (e.g. biomass burning from South America), although differences in the chemical schemes may also be influential and would require further analysis.

Differences with AKs show that EMAC is in slightly better agreement with OMI across the Southern Hemisphere extratropics, although CMAM is in closer or comparable agreement over the tropics and the Northern Hemisphere. The

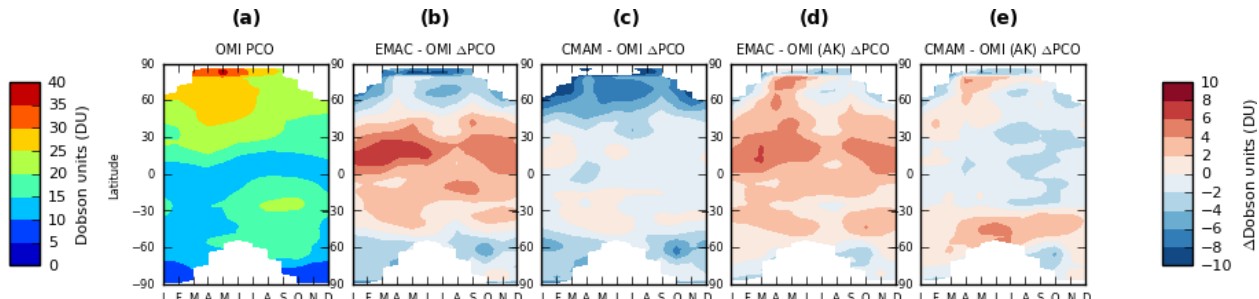

**Figure 2.** Zonal-mean monthly averaged 1000–450 hPa (0–5.5 km) sub-column O$_3$ (DU) for 2005–2010 from **(a)** OMI, **(b)** EMAC minus OMI without AKs, **(c)** CMAM minus OMI without AKs, **(d)** EMAC minus OMI with AKs and **(e)** CMAM minus OMI with AKs.

model is especially consistent during JJA and SON over the continents in particular (Fig. 1b and c). Furthermore, a high level of agreement between the ozone-sonde and OMI observations is apparent in all four seasons (Fig. 1a), confirming that the OMI retrieval algorithm correctly captures the regional and seasonal climatological features in tropospheric ozone. Some sonde sites, however, show consistently smaller amounts of ozone (e.g. western North America and Greenland), although this may be attributed to the high elevation (e.g. mountain summit locations) of these sites relative to the average topographical elevation of a 2.5° grid cell within which the OMI observations are averaged, which inherently leads to lower amounts of ozone within the partial column.

### 3.1 O$_3$ inter-annual variability

As a metric of inter-annual variability, seasonal aggregates of the computed relative standard deviation (RSD) of the monthly mean ozone for OMI, each CCM and ozone-sondes are shown in Fig. 3, as calculated in Eq. (1) below:

$$\text{RSD} = \sum_{i=0}^{N} \frac{\sigma_i}{\mu_i} \, / \, N, \tag{1}$$

**TS4** where $N$ is the number of months in a season, $\sigma_i$ is the standard deviation of each month calculated over all years and $\mu_i$ is the multiannual monthly mean of each month. Variability in the tropics is enhanced due to the significantly lower mean tropospheric ozone, in comparison with the extratropics. It should be noted that the calculated RSD is significantly lower for ozone-sondes compared to each CCM and particularly the OMI measurements, which is currently being investigated further.

Although OMI shows much higher variability than the models, there is good agreement in regions of high RSD across much of the tropics ($> 10\,\%$), which is largest during SON, at least from the OMI observations. The highest RSD is consistently found over the western Pacific and the Maritime Continent close to the Equator, where it approaches 20 % for both OMI and the CCMs (particularly CMAM). The region is strongly influenced by some of the main drivers

of natural variability, including the El Niño–Southern Oscillation (ENSO) and the Madden–Julian Oscillation (MJO). Throughout the tropics, high variability may also be associated with the QBO. Although the QBO is a stratospheric phenomenon, studies show that the alternating phases of the zonal equatorial wind can influence tropospheric ozone by as much as 10 %–20 % ($\sim 8$ ppbv) (e.g. Lee et al., 2010). The RSD is generally lower for OMI outside of the tropics, although significant variability ($> 10\,\%$) is still evident for some regions in different seasons. The CCMs in contrast show very low RSD over much of the extratropics ($< 5\,\%$), with only a subtle spatial structure evident in the seasonal composites. Equivalent composites of the absolute standard deviation (not shown) show some variability, however, at mid-latitudes during winter and spring in each hemisphere (up to 2 DU), principally in oceanic regions, and this may indicate sensitivity to the main extratropical cyclone tracks. Higher RSD is, however, shown across Antarctica during the polar day and over the Southern Ocean (up to 10 %), which is collocated in the corresponding OMI seasonal composites. This may largely be a retrieval artefact caused by vertical smearing, which is highly dependent on the tropopause height, since comparative RSD fields from the CCMs without AKs show no such structure (not shown).

### 3.2 O$_3$ vertical distribution assessment

To evaluate the vertical agreement of the CCM O$_3$ VMR tracer simulations, monthly mean ozone-sonde-derived measurements were interpolated and averaged between $\pm 20$ hPa of the 22 different model pressure levels between the surface ($\sim 1000$ hPa) and the lower stratosphere (100 hPa) for three different extratropical regions. Figure 4 shows the monthly mean evolution averaged over all sites (left), together with the respective percentage differences relative to the nearest model grid columns in EMAC (middle) and CMAM (right), within each bounding box (region): (a) Europe (30–65° N, 15° W–35° E), (b) eastern North America (32.5–60° N, 92.5–55° W) and (c) the Tasman Sea (55–15° S, 140–180° E). The absolute differences are also shown in Fig. S1. Table S2a–c additionally provide a summary of this information on a

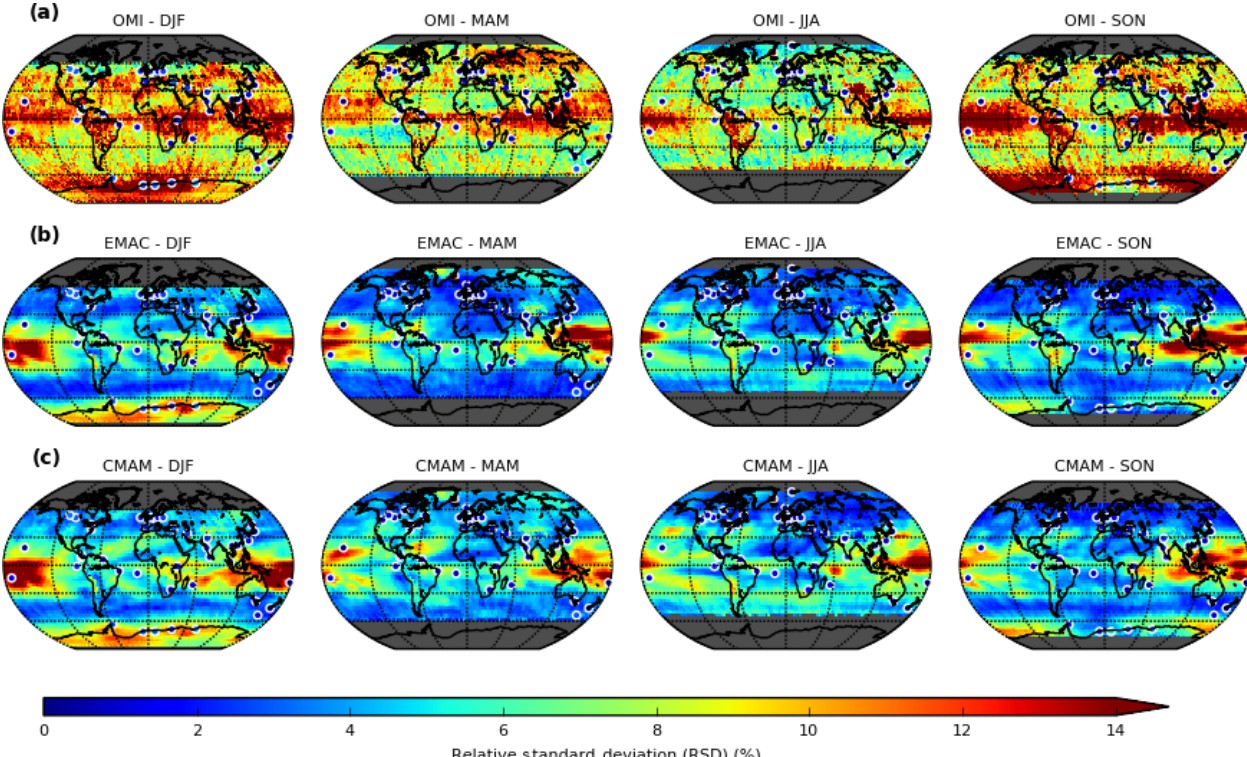

**Figure 3.** Seasonal composites (left to right) of monthly 1000–450 hPa (0–5.5 km) sub-column O$_3$ relative standard deviation (RSD) (%) for 2005–2010 for **(a)** OMI, **(b)** EMAC and **(c)** CMAM. Circles denote **(a–c)** the seasonal RSD calculated from ozone-sonde measurements. Model and ozone-sonde sub-column data have again been modified using AKs to ensure a direct comparison.

seasonal basis for six selected pressure levels in each region. These regions were selected for the assessment due to the relatively high number of ozone-sonde sites in close proximity. Furthermore, the variability in emissions of ozone precursors and stratospheric influence, due to varying upper-troposphere–lower-stratosphere (UTLS) dynamics in these predominantly extratropical regions, make these regions suitable for evaluating the realism to which the CCMs simulate these influences.

The seasonality in ozone VMR is shown to be very similar in both Europe (Fig. 4a) and eastern North America (Fig. 4b) as expected for two regions of similar latitude in the same hemisphere. In the stratosphere, a springtime maximum (autumn minimum) is clear, although the timing is not synchronous at all pressure levels, with a tendency for a delayed maximum (minimum) in each region with increasing pressure (decreasing altitude). This is also apparent for the Tasman Sea region (Fig. 4c), albeit with a reversed seasonality. This can be attributed to the BDC in the lower stratosphere, which leads to a gradual accumulation of ozone during wintertime in the lowermost stratosphere and a subsequent gradual depletion of ozone during summertime as the circulation weakens (Logan, 1985; Holton et al., 1995; Hegglin et al., 2006). For all regions, this delayed signal in the maximum (minimum) in ozone VMR propagates down into the tropo-

sphere (identified here as the region < 100 ppbv), with the exception of the springtime maximum over the Tasman Sea, which peaks earlier with increasing pressure (decreasing altitude) from the tropopause (around late September) towards the surface (early August). Clearly though, there is a large difference in the climatological ozone VMR throughout the year between this region and both Europe and eastern North America; the Tasman Sea region reflecting only a very limited influence from emission precursors. The composite produced for this region likely provides a reasonable representation of the natural background influence of the stratosphere on tropospheric ozone in the extratropics, in contrast to the other two regions.

The computed model–ozone-sonde monthly mean differences (Fig. 4) reveal notable differences both between each model and each region in the troposphere ($\sim 300$–1000 hPa) as high as 20–30 ppbv (> 50 %). EMAC shows an almost universal positive bias between 0 % and 40 % (0–20 ppbv) throughout the year for all three regions, which contrasts with the overall negative bias in the stratosphere ($\sim 100$–300 hPa) (except over eastern North America). Some seasonal dependence in the tropospheric bias is evident over Europe and eastern North America, with the largest (smallest) difference between September and May (June and August), on the order of $\sim +20$ %–60 % (+10–20 ppbv) outside of boreal summer.

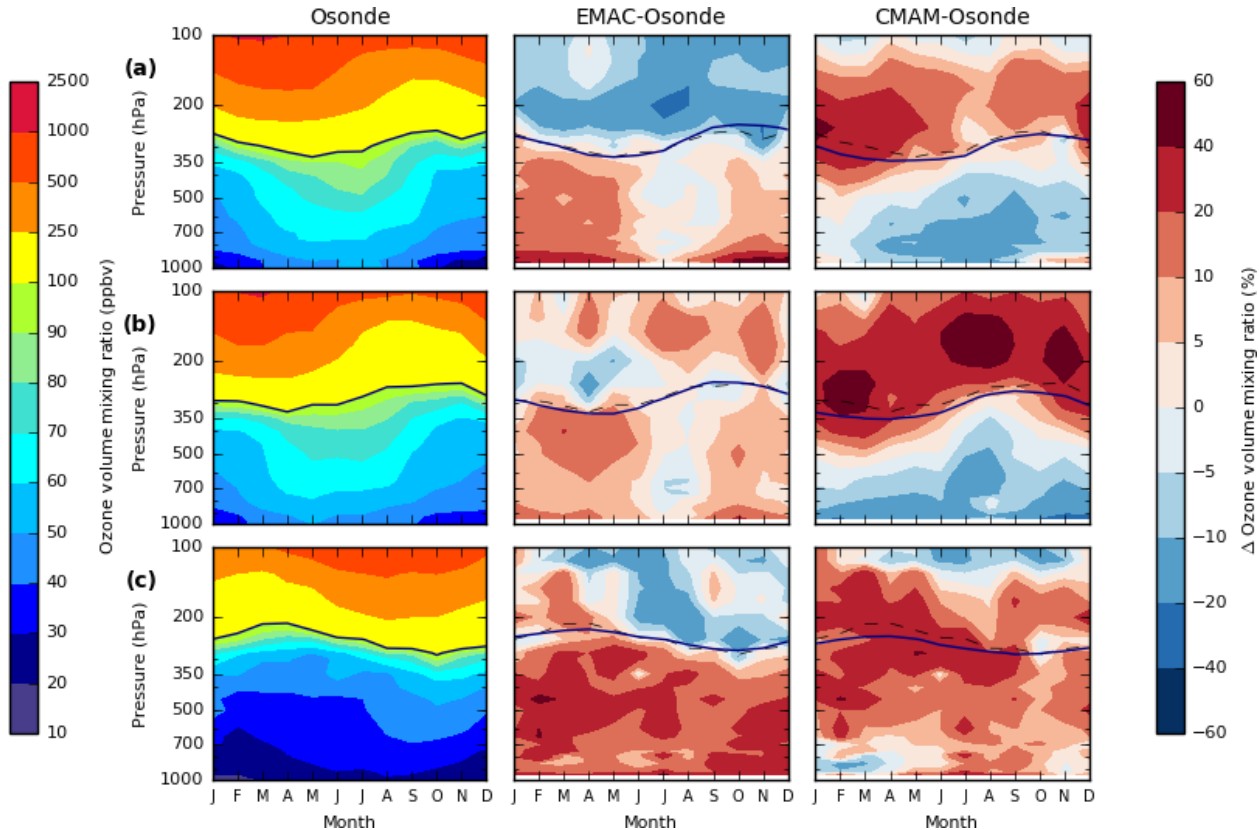

**Figure 4.** Monthly evolution of the vertical distribution of mean $O_3$ VMR (ppbv) derived from ozone-sonde measurements (left column); EMAC minus ozone-sonde differences (%) (middle column) and CMAM minus ozone-sonde differences (%) (right column) over the period 1980–2010 inclusive for three different world regions: **(a)** Europe ($n = 18$), **(b)** eastern North America ($n = 14$) and **(c)** the Tasman Sea ($n = 6$). The ozone-sonde–model 100 ppbv contour is additionally highlighted in bold (ozone-sonde 100 ppbv contour indicated again by dashed line – middle and right column).

In contrast, no obvious seasonal variation in the bias is apparent over the Tasman Sea region. For CMAM, a generally negative, seasonally dependent bias ($\sim$ 5 %–20 % or 5–10 ppbv) is apparent in the lower to middle troposphere over Europe and particularly eastern North America, most pronounced during summer (JJA), whereas an overall positive bias (up to 10 %–40 % or 5–20 ppbv) exists over the Tasman Sea and is largest in the free troposphere. Both the seasonal character of the negative bias over Europe and eastern North America (largest during the most photochemically active months), together with the difference in the sign of the bias between the troposphere and the UTLS, strongly implies a difference in the implementation of the tropospheric chemistry scheme in CMAM compared with EMAC, since prescribed emissions are equivalent in both models. Specifically, the omission of non-methane VOCs (NMVOCs) in CMAM likely accounts for much of this underestimation.

The largest absolute differences (Fig. S1 in the Supplement) are, however, indeed evident in the lower stratosphere (100–300 hPa), with a systematic positive bias in CMAM in most seasons (widely between +50 and +200 ppbv, rang-

ing from 10 % to 50 %). A slight negative bias ($\sim$ −10 to −50 ppbv or −0 % to 10 % TS5) is, however, apparent between 100 and 150 hPa over Europe, largely during summer (JJA), and is also more pronounced over the Tasman Sea from March through to November (> 50 ppbv or 5 %–20 % TS6). Over eastern North America, a very large positive bias is evident in CMAM throughout the year ranging between 20 % and 60 % (+50 and +200 ppbv), with a seasonal shift in the height of the largest differences, similarly to over Europe yet more pronounced. In contrast, the differences between EMAC and the ozone-sonde measurements have a very different character, with a general negative bias over Europe, particularly in summer (JJA) ($\sim$ −20 to −100 ppbv or −10 % to 20 % TS7). Over eastern North America and the Tasman Sea, the pattern and magnitude of the biases is more complex with both pressure (altitude) and month. An overall positive bias is found over eastern North America (typically +20 to +50 ppbv or +5 %–20 %), except from January to May between $\sim$ 170 and 250 hPa, whilst an overall negative bias (generally between −20 and −50 ppbv or 5 % and 20 % TS8) is evident over the Tasman Sea except between

Please note the remarks at the end of the manuscript.

January and May and for a small region (120–180 hPa) during August–September. The general negative bias in EMAC (positive bias in CMAM) might indicate an underestimation (overestimation) in the strength of the BDC, but the seasonal dependence of the bias, and in particular the complexity in EMAC, suggests influence from other factors.

### 3.3  Summary

In summary, the CCM simulations are broadly in agreement with both sets of observations, capturing both the extent and magnitude of geographical and seasonal features in tropospheric ozone over the concurrent period of data availability (2005–2010). There is very close agreement overall in the global mean seasonal composites of tropospheric sub-column (1000–450 hPa) ozone between both CCMs, although differences relative to OMI show that there is an overall significant, systematic positive bias in the EMAC model (Figs. 1 and 2), particularly over the Northern Hemisphere ($\sim$ 2–8 DU), whereas no overall bias is apparent in CMAM despite some meridional and seasonal differences ($\sim$ −4 to +4 DU). An evaluation of the model–ozone-sonde differences in the vertical distribution of ozone VMRs (ppbv) over both Europe and eastern North America (Fig. 4) indicates a different origin for the biases in each model compared with OMI. In EMAC, the positive bias is predominantly a result of an excess of in situ photochemical production from emission precursors, whereas biases in CMAM are largely determined by the relative influence of excessive vertical smearing of ozone (induced by applying the OMI AKs). This results from a large positive ozone bias in the lower stratosphere (not present in EMAC) as well as the much more simplified tropospheric chemistry scheme implementation. The additional smearing from AKs is concluded to overcompensate for the reduced in situ production of ozone to yield a larger positive or comparable bias in CMAM (poleward of 30° S/N) (Fig. 2), where the application of AKs has a disproportionally larger effect on the estimated sub-columns. In contrast, a larger positive bias is found in EMAC over low latitudes (30° N–30° S) but primarily in the Northern Hemisphere where precursor emissions are more abundant, which is understandable due to the higher climatological mean position of the tropopause in this region (with respect to the extratropics), leading to less vertical smearing of information from the stratosphere when AKs are applied. The zonal average monthly mean integrated sub-column OMI–model differences without AKs (Fig. 2b–c) would be consistent with this interpretation and it is obvious that application of the OMI AKs must have induced additional vertical smearing of ozone in CMAM in the equivalent latitude range ($\sim$ 30–65° N) compared with EMAC (Fig. 2d–e) due to the likely presence of a high ozone bias in the lower stratosphere compared with both ozone-sondes and EMAC. Such a factor is also suspected to be influential in explaining the transition from a negative to a positive bias after applying AKs in the Southern Hemisphere

between May and December in the region between 30 and 60° S in CMAM. The sensitivity of the 1000–450 hPa sub-column to the lowermost stratosphere is exemplified in a plot of the monthly mean AKs for August 2007 over the Southern Ocean ($\sim$ 47° S, 0° E) (Fig. S2), which shows influence from the $\sim$ 150–450 hPa pressure range. It is known that CCMs tend to have inherent biases in ozone in the lower stratosphere (e.g. Jöckel et al., 2006, 2016; Pendlebury et al., 2015; Kolonjari et al., 2018), so it is likely that the results found here are applicable hemisphere-wide but again further investigation is warranted, perhaps using an ozone-sonde trajectory-based mapping approach (e.g. Liu et al., 2013). The interannual variability (Fig. 3) in the models seems to be consistent with that from the OMI measurements and as reported in the literature, at least in the equatorial region where the magnitude of interannual variability is typically on the order of $\sim$ 10 %–20 %. In the extratropics, both ozone-sondes and models show smaller variability (< 5 %), in contrast to OMI. Whether such differences arise due to model inadequacies in capturing the magnitude of natural variability or simply as a result of measurement noise in the OMI observations is a subject for further investigation.

## 4  Stratospheric influence

Having assessed the ability of the CCMs to represent key features of the global climatology of tropospheric ozone with respect to both in situ and satellite observations, model simulations of the vertical distribution in ozone VMR are now investigated globally over the 1980–2010 climatological period, together with the role of stratospheric ozone in influencing both regional and seasonal variations.

### 4.1  O$_3$ vertical distribution, seasonality and stratospheric contribution (O$_3$F)

Seasonal composites of the monthly mean, zonal-mean vertical distribution of ozone VMR in the troposphere and lower stratosphere (1000–80 hPa) are shown in Fig. 5 for (a) EMAC, (b) CMAM and (c) CMAM-EMAC, together with the percentage contribution of mean ozone of stratospheric origin (O$_3$F (%) = (O$_3$S/O$_3$) $\times$ 100: dashed lines). The equivalent seasonal composites of tagged stratospheric ozone (O$_3$S) VMR are also shown in Fig. S3. The meridional distribution in the tropospheric seasonal mean ozone VMR corresponds closely to the latitudinal variability in the integrated 1000–450 hPa sub-column seasonal composites produced from both the CCM and OMI data (Figs. 1 and 2). The highest ozone VMR according to both CCMs can be found over mid-latitudes, with consistent seasonality to that identified in Sect. 3 and a maximum in the Northern Hemisphere during spring into summer (MAM and JJA) and in spring (SON) over the Southern Hemisphere. It is obvious that ozone VMR is significantly greater year-round in the

Northern Hemisphere. This is due in part to the large difference in precursor emissions from the surface but also due to a stronger BDC in the Northern Hemisphere and a subsequent enhanced STE of ozone, with the former clearly a greater influence near the surface and the latter in the upper troposphere. As indicated by the dashed contours, the stratospheric influence increases with altitude for all latitudes across all seasons. However, there is a significant meridional gradient in the stratospheric influence, with values ranging from < 30 % over the tropics in all four seasons throughout the troposphere to maximum values between 40 % and 75 % during the winter months at high latitudes in both hemispheres from the surface to 350 hPa. Towards summertime, this fraction decreases sharply across middle and high latitudes (particularly near the surface) due to a combination of reduced STE and increasing importance of precursor emissions during the photochemically active months. Thus in relative terms, the stratosphere has a smaller contribution outside the winter months (lowest in summer). Despite this, the stratosphere has the largest contribution during spring in absolute terms (see Supplement Fig. S3), extended through to summer in the Northern Hemisphere upper troposphere, as is well established in the literature (e.g. Richards et al., 2013; Škerlak et al., 2014; Zanis et al., 2014). This further implies that the influence of the stratosphere becomes secondary to precursor emissions during the photochemically active months, away from the upper troposphere.

The inter-model difference in the zonal mean ozone VMR for each season is shown in Fig. 5c. With respect to EMAC, CMAM shows lower values overall throughout the tropical troposphere and also over the Northern Hemisphere lower and middle troposphere in all seasons ($\sim 0\,\%$–$30\,\%$ or between 0 and $-20$ ppbv). In contrast, CMAM shows much higher values in the extratropical upper troposphere (up to $+50$ ppbv or $50\,\%$–$100\,\%$ in relative terms) in all seasons, with smaller positive differences extending towards the surface in the Southern Hemisphere, particularly in winter (JJA). The large difference in the extratropical upper troposphere, in conjunction with the vertically extensive negative bias in the tropics, may be partially attributed to a difference in the large-scale dynamics in each model. Notably, a modest downward shift in the height of the extratropical tropopause would lead to such large differences apparent in Fig. 5, due to the existence of a very sharp gradient in ozone VMR at this boundary. Indeed, it has been identified previously that tropopause pressures in EMAC are lower than CMAM (by as much as 30–50 hPa) in free-running simulations, equating to a smaller total mass of the lowermost stratosphere (Hegglin et al., 2010), although the actual difference is likely smaller in the case of the specified-dynamics simulations analysed here. Apart from over the Southern Hemisphere high latitudes, the negative difference in CMAM (relative to EMAC) throughout much of the troposphere would appear to be related to both a difference in the implementation of the tropospheric chemistry scheme in each model and the amount of simulated $O_3S$, which is evidently some 0–10 ppbv (up to 20 %) lower in CMAM despite a much larger ozone burden in the extratropical UTLS region (Fig. S3c). An exception to this is over the Southern Hemisphere subtropics during wintertime (JJA), especially where a significantly larger amount of $O_3S$ ($\sim 0\,\%$–$20\,\%$) is transported down towards the surface in CMAM compared with EMAC (indicative of greater STE). The absence of a positive difference in Fig. 5c in this region, however, suggests an overwhelming influence of the reduced in situ photochemical formation of ozone in CMAM due to the simplicity of the tropospheric chemistry scheme in this model, despite an obvious larger stratospheric ozone fraction here ($O_3F > 20\,\%$ larger in CMAM in the mid-troposphere).

## 4.2 $O_3F$ global distribution and seasonality

The global distribution of ozone of stratospheric origin is next investigated in order to quantify the relative contribution to tropospheric ozone as well to help identify preferential pathways of stratosphere–troposphere transport. The climatological fraction of stratospherically sourced ozone ($O_3F$) is shown globally for EMAC and CMAM, together with the difference between both models (CMAM-EMAC) in Fig. 6 at (a) 350 hPa, (b) 500 hPa and (c) 850 hPa for both DJF and JJA (see Fig. S4 for MAM and SON) over the period 1980–2010, when $O_3F$ reaches a maximum in winter and minimum in summer. Both CCMs are broadly consistent at each pressure level, with a clear decrease in the $O_3F$ towards the surface as already indicated in Fig. 5. The meridional gradient is largest in the upper troposphere at 350 hPa, with low values across the tropics ($< 40\,\%$ between $30°$ N and $30°$ S) associated with both convective upwelling and the short photochemical lifetime of ozone in the tropics and with higher values in the extratropics but particularly in the winter hemisphere ($> 70\,\%$). In the Northern Hemisphere mid-latitudes, where the gradient is largest, a planetary-scale wave pattern is evident (particularly at 350 hPa), which is consistent with longitudinal variability in the climatological positioning of the upper-level jet streams induced by orography (e.g. the Rocky Mountains in North America) (Charney and Eliassen, 1949; Bolin, 1950), particularly in winter (DJF). Although the $O_3F$ is relatively high during summer in each hemisphere at 350 hPa as well, the $O_3F$ is much lower at 500 and 850 hPa (which is consistent with Fig. 5) and reflects the relatively minimal role of the stratosphere during this season (with strong influence from precursor emissions instead). At 850 hPa, the stratospheric influence is typically largest over oceanic regions, which further reflects the importance of emission precursors over continental regions, particularly in the Southern Hemisphere where biomass burning is prevalent over Africa and South America.

Large differences in $O_3F$ are apparent at high latitudes (poleward of $60°$ N and $60°$ S) during summer in each hemisphere at 350 hPa, with CMAM showing a significantly

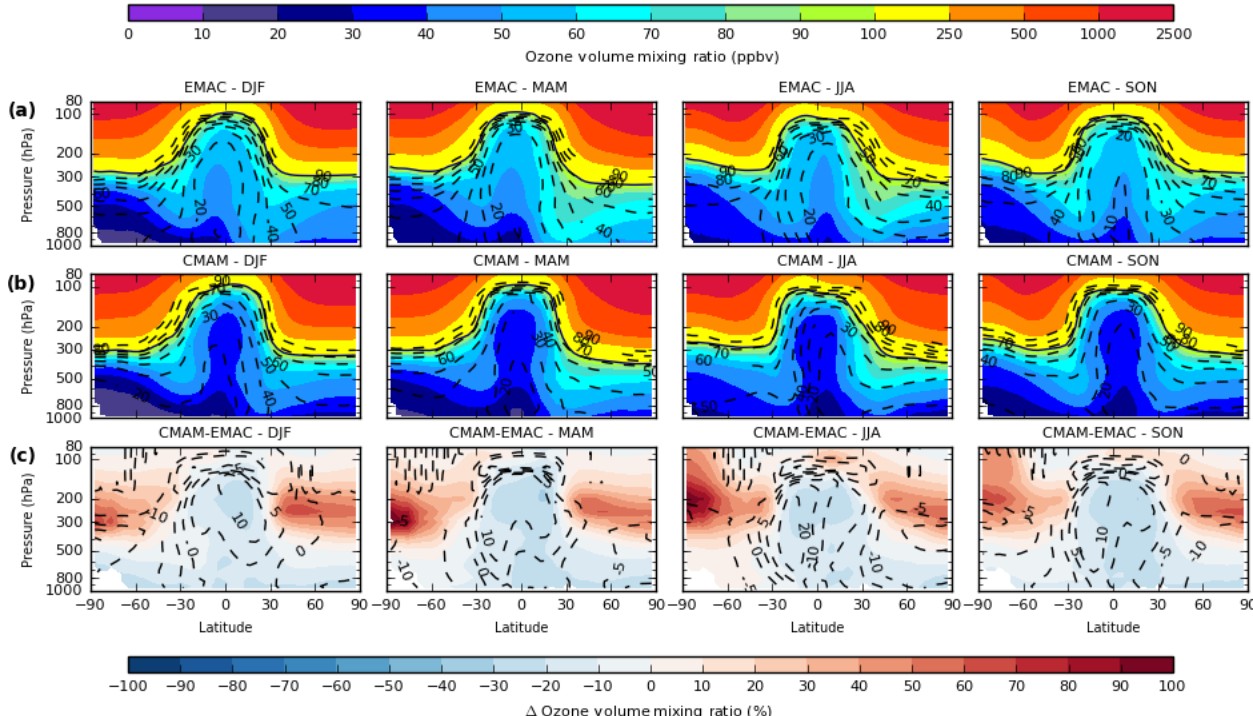

**Figure 5.** Zonal mean seasonal composites of monthly mean $O_3$ VMR (ppbv) for the troposphere and lower stratosphere (1000–80 hPa) from **(a)** EMAC, **(b)** CMAM, and **(c)** CMAM and EMAC (CMAM-EMAC) percentage differences over the period 1980–2010. Dashed lines indicate the stratospheric contribution (%) calculated using both ozone tracers in each model: $O_3F$ (%) = $(O_3S/O_3)$ ×100. The 100 ppbv contour (bold line) is included as a reference for the tropopause altitude **(a–b)**.

smaller fraction in the ozone of stratospheric origin ($\sim$ 40 %–50 %) compared with EMAC ($\sim$ 70 %–80 %). This is despite a positive bias of $\sim$ 20 %–50 % (20–30 ppbv) in the seasonal mean ozone VMR in CMAM compared with EMAC (Fig. 5c), although this bias exists across all seasons whereas the $O_3F$ bias is seasonally dependent. Inspection of model tracer values (not shown) indicates slightly lower-stratospheric ozone ($O_3S$) in CMAM compared with EMAC, along with higher $O_3$ values (ozone of non-stratospheric origin) at 350 hPa, which gives rise to this difference although the exact origin of this discrepancy would require further investigation. During wintertime in the Southern Hemisphere (JJA) subtropics, a large positive difference in $O_3F$ also exists over a relatively narrow latitude range between 0 and 30° S, which is indicative of an equatorward displacement in the position of the subtropical jet stream in CMAM compared with EMAC. The differences show some variation longitudinally, with the largest differences extending from east Africa towards Indonesia and northern Australia and out across the South Pacific. Reference to seasonal composites of the model $O_3S$ VMR tracer (Fig. S3) confirms that the positive bias is related to larger STE in CMAM relative to EMAC, at least over the Southern Hemisphere subtropics. The effect of greater STE, even locally across this latitude range, in CMAM would propagate eastwards due to the influence of upper-level winds, leading to the transport of ozone-

rich air on intercontinental scales. Both the highest $O_3S$ (not shown) and $O_3F$ values in CMAM are apparent over a relatively small geographical area of the Indian Ocean north of Madagascar (adjacent to the east African coastline), which signifies preferential stratosphere-to-troposphere transport in this region which extends deep into the lower troposphere ($O_3F > 50$ % at 850 hPa). Although EMAC shows relatively high $O_3F$ in the wider region during this season, evidence of a preferential STE pathway here is lacking in this model, and indeed no such feature has been widely recognised in the literature. Such differences are non-existent during DJF, although CMAM shows generally higher $O_3F$ over part of the Indian Ocean and the South Pacific and relatively lower $O_3F$ over South America, the South Atlantic and over Africa. The differences described at 350 hPa are very similar at 500 hPa, albeit with a lower negative difference at high latitudes during summer ($\sim$ 10 %–20 %). Although the spatial distribution of the biases is broadly consistent at 850 hPa as well, there is much greater variability regionally in the tropics and the negative bias at high latitudes is relatively low ($> 10$ %).

### 4.3 Monthly evolution of stratospheric influence

The zonal-mean monthly evolution of mean ozone ($O_3$) VMR at 350, 500 and 850 hPa is shown in Fig. 7 (a) based on the monthly mean aggregated in situ ozone-sonde observa-

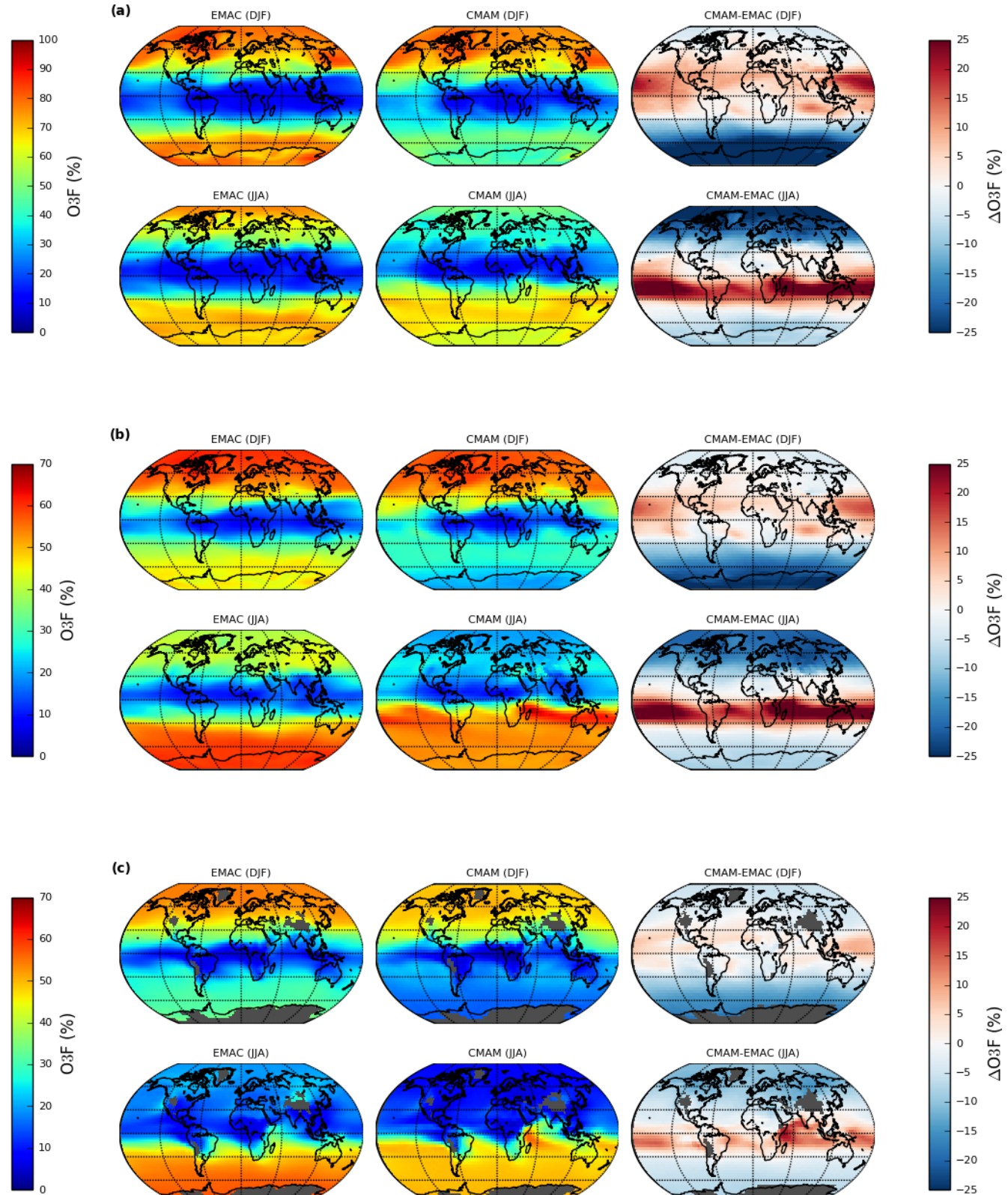

**Figure 6.** Seasonal (DJF and JJA) composites of **(a)** 350 hPa, **(b)** 500 hPa and **(c)** 850 hPa monthly mean stratospheric ozone fraction ($O_3F$) for EMAC (left), CMAM (middle) and CMAM-EMAC (right) over the period 1980–2010. Note the scale difference between **(a)** and **(b–c)**. Grey shaded regions represent regions where the surface pressure is lower than the plotted pressure level.

tions from the WOUDC database, interpolated and averaged for $10°$ latitude intervals and within $\pm 20$ hPa of each pressure level, (b) as simulated by EMAC, and subsequently (c) for EMAC $O_3S$ and (d) EMAC $O_3F$. The ozone-sonde measurements are in broad agreement with that simulated by EMAC (and CMAM; see Fig. S5) in terms of both the seasonality and meridional variability in the climatological mean ozone VMR at each of the three different pressure levels. However, the ozone VMR across the Northern Hemisphere high latitudes at both 500 and 850 hPa during the broad spring and summer maximum is somewhat higher ($\sim 0$–10 ppbv) in EMAC, whereas closer agreement with the ozone-sonde climatology is apparent for CMAM (Fig. S5). At the 350 hPa level on the other hand, CMAM overestimates ozone in the extratropics relative to both EMAC and ozone-sondes by as much as 10–20 ppbv, which is consistent with the identified high ozone bias in the UTLS in CMAM over three different extratropical regions in Sect. 3 (Fig. 4), whereas EMAC is in closer agreement with the ozone-sonde-derived composites. Furthermore, there is very high variability with latitude in the tropics compared with EMAC (and CMAM), although this is almost certainly an artefact of both the paucity and poor spatial representativeness of ozone-sonde stations. This figure is similar to that produced by Lamarque et al. (1999, Fig. 2., p. 26 368) and their model results bear some resemblance to Fig. 7 (Fig. S5) in terms of the characterisation of the zonal mean evolution of ozone VMR and calculated $O_3F$, although significantly higher $O_3$ and $O_3S$ VMRs are evident in the CCM simulations as are higher stratospheric fraction ($O_3F$) values in this study.

The EMAC $O_3S$ evolution corresponds closely to the $O_3$ evolution at 350 hPa, reflecting the large contribution of the stratosphere in the upper-troposphere ozone burden (shown also in the $O_3F$ evolution), but this correspondence falls sharply towards the surface (850 hPa) as noted in Sect. 4.2 from Fig. 6. It is important to note that a pronounced spring maximum in $O_3S$ ($> 60$ ppbv at 350 hPa) is only evident in the Northern Hemisphere, with a much smaller, short-lived maximum between 30 and $60°$ S ($\sim 40$ ppbv at 350 hPa), due to the combined influence of the springtime Antarctic ozone hole and a weaker BDC in the Southern Hemisphere which constrains the seasonality. The ozone hole influence is particularly apparent at 350 hPa in each model $O_3F$ evolution fields (Fig. 7d), where the strong symmetry between each hemisphere is briefly interrupted during SON when the ozone hole readily develops over the Southern Hemisphere high latitudes. The $O_3F$ evolution again shows the sharp meridional gradient in the stratospheric influence, particularly in the upper troposphere, which separates the tropical zone of convective upwelling from the region of net subsidence in both hemispheres where net STE is downward. The seasonality in extratropical $O_3F$ is greater towards the surface due to the competing influence of precursor emissions. Despite this, Fig. 7 (bottom row) shows that the stratosphere still contributes about half ($\sim 50$ %) of the amount of ozone during winter at high latitudes at 850 hPa, implying that the stratosphere has a significant influence on near-surface ozone levels and, in turn, air quality. This fraction is slightly higher in the Southern Hemisphere due to the lower abundance of precursors compared with the Northern Hemisphere.

## 4.4 Summary

In summary of this section, the use of the model stratospheric ozone ($O_3S$) tracers reveals a significant difference in the strength and dominance of the shallow branch of the BDC in each model, which is intrinsically related to the burden of ozone in the extratropical lowermost stratosphere through transport from the primary ozone production (equatorial) region (Hegglin et al., 2006). This has implications for both the simulated downward flux of ozone from the stratosphere and its influence on the relative contribution of stratospheric ozone to tropospheric ozone. CMAM simulates a faster, shallower BDC as inferred from Fig. 5 (Sect. 4.1), which shows between 50 %–100 % more ozone in the extratropical UTLS region (equating to as much as a +50 ppbv difference), which contrasts with a negative difference in the tropics of between 0 % and 30 % (0 and −20 ppbv difference) relative to EMAC within this region ($\sim 200$–400 hPa). This inference is supported by a recent finding of a maximum decrease in the age of air (AoA) between 1970 and 2100 in the mid-latitude lower stratosphere in CMAM, whereas EMAC shows a decrease in stratospheric mean AoA which is more pronounced with both latitude and altitude, due to the acceleration of the BDC due to climate change (Eichinger et al., 2019). It is inferred from a characterisation of the vertical ozone distribution biases in Fig. 4 (Sect. 3) that EMAC more accurately depicts the BDC and its effects on the meridional variation in stratospheric ozone, although it is likely that this model is still too conservative in this aspect compared to reality, given a smaller, but general negative stratospheric ozone bias (up to 10 %–20 %) in the extratropics with respect to ozone-sondes. The same inference is in turn made for the STE of ozone; a larger proportion of the downward flux of ozone is simulated over the subtropics in comparison with EMAC, which simulates a larger flux in the extratropics (Figs. 6 and S4). The difference is particularly large in the Southern Hemisphere subtropics (0–$30°$ S), with a typically larger fraction of stratospheric ozone ranging from 10 % to 25 % from the lower to upper troposphere in CMAM relative to EMAC during austral winter (JJA). There is an indication of a preferential STE pathway over the western Indian Ocean and neighbouring east Africa which is active during this season as far down as the PBL according to CMAM, although any preferential pathway or STE "hotspot" in this region is neither obvious in EMAC nor widely established in the literature. Further work is necessary to understand how realistic the representation of STE is in each model, together with the simulated in situ photochemical production of ozone from precursor emissions. Reference to the earlier work of Lamar-

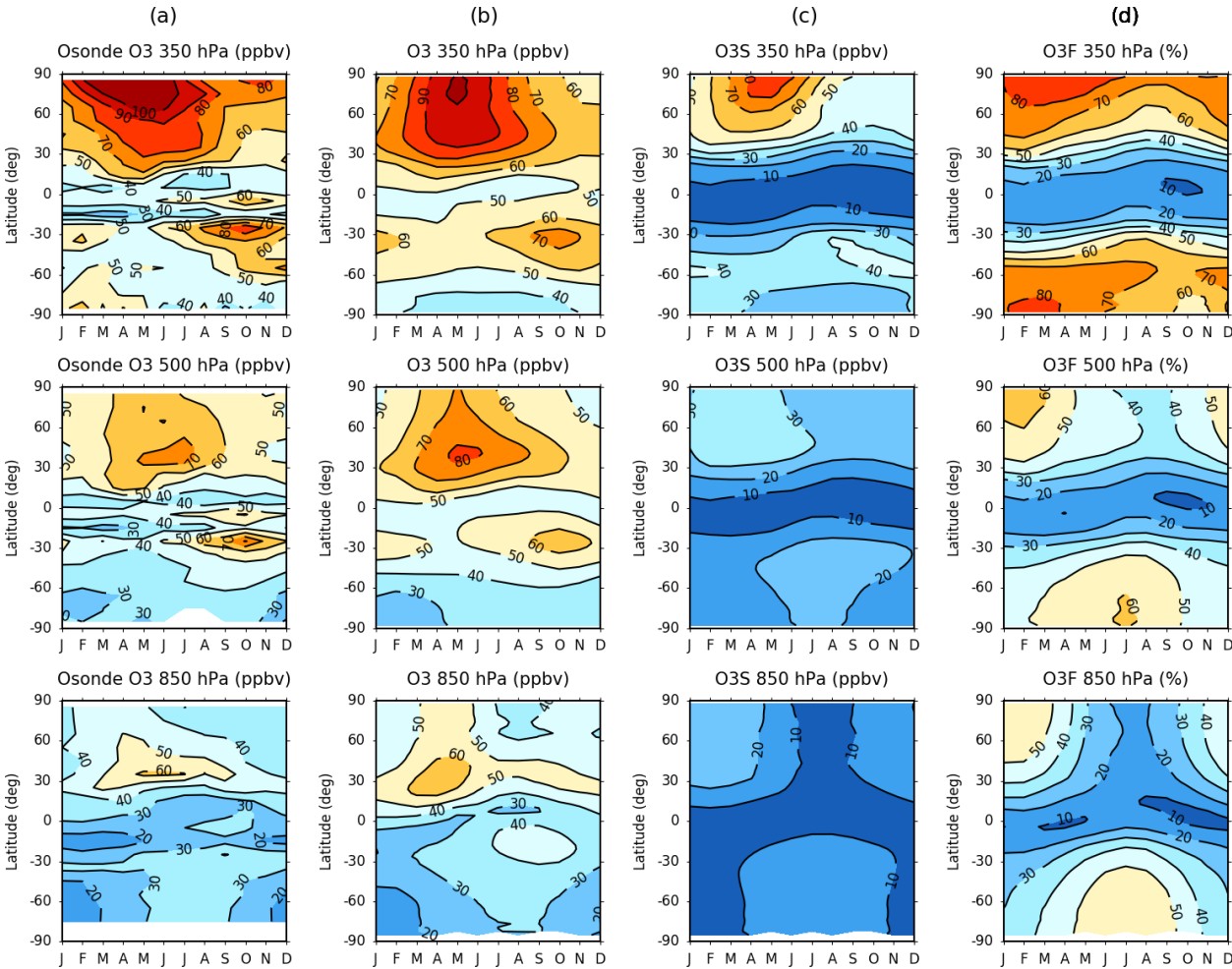

**Figure 7.** Zonal-mean monthly mean evolution of $O_3$ VMR (ppbv) derived from **(a)** ozone-sondes and **(b)** EMAC $O_3$ model tracer. The evolution of the **(c)** EMAC stratospheric $O_3S$ tracer and **(d)** $O_3F$ stratospheric fraction (%) are additionally included over the period 1980–2010 for 350 hPa (top row), 500 hPa (middle row) and 850 hPa (bottom row).

que et al. (1999) shows that the contemporary CCM simulations analysed in this study more closely match the ozone-sonde-derived climatology, which is remarkably consistent in both this study and that produced by Lamarque et al. (1999, Fig. 1, p. 26 367), compared to the chemistry transport model (CTM) selected in their study, which underestimated tropospheric ozone VMRs by as much as 20 %–50 %. Both the stratospheric ozone and derived stratospheric fraction fields in their study show very conservative numbers relative to that calculated in this study for both EMAC and CMAM, indicating that the stratosphere has a much larger influence than previously thought, although differences in the stratospheric tracer definitions might explain some of this difference. Both contemporary simulations suggest a significant stratospheric influence on tropospheric ozone of over 50 % during wintertime in the extratropics (extending down into the lower troposphere), which is significantly higher than the 10 %–20 % estimated from the CTM in Lamarque et al. (1999) and still

considerably higher than more recent studies, which imply an influence in the range of 30 %–50 % (e.g. Lelieveld and Dentener, 2000; Banerjee et al., 2016).

## 5 Recent changes in tropospheric $O_3$ and $O_3S$

Seasonal changes in the global mean tropospheric ozone distribution between 1980–1989 and 2001–2010 are next quantified using the CCM simulations, together with changes in attribution from the stratosphere. The changes in the simulated ozone ($O_3$) VMRs between these two periods are shown globally in Fig. 8 at 350, 500 hPa and the surface model level as well as throughout the troposphere for three different latitudinal cross sections (30° W, 30° E and 90° E) in Fig. 9 for MAM and SON. Changes for DJF and JJA are also shown globally in Fig. S6 and for these latitude cross sections in Fig. S7. These latitudinal transects help show that regional changes in $O_3$ and $O_3S$ are strongly height-dependent, partic-

ularly along these selected longitudes where notable features are observed, which differ in each model and season. The respective changes in the simulated stratospheric ozone ($O_3S$) VMRs are then shown globally in Fig. 10 (Fig. S8) for each level and as a function of pressure for each latitudinal cross section in Fig. 11 for MAM and SON and Fig. S9 for DJF and JJA. Zonal-mean changes in each model tracer are additionally summarised in Table S3 ($O_3$) and Table S4 ($O_3S$) for 30° latitude bands. Statistical significance is inferred where the paired $t$ test $p$ value is less than 0.05 (stippled regions), although the distribution of such regions should be interpreted only as an approximation in the absence of additional data (Wasserstein and Lazar, 2016).

## 5.1 $O_3$ change (1980–1989 to 2001–2010)

It is evident in Fig. 8 that both models simulate an overall increase in ozone, which is typically largest (in absolute terms) and most robust (statistically significant) in the upper troposphere (350 hPa) and across the Northern Hemisphere in both seasons. The increase here in both MAM and SON is on the order of some 4–6 ppbv (5 %–10 %), although in excess of 6 ppbv across some regions during MAM and in CMAM especially, with only a slightly smaller overall increase evident at 500 hPa (mid-troposphere). Greater spatial variability is evident at 350 hPa (at least in MAM) due to enhanced sensitivity to changes in the tropopause altitude at this level. This can be inferred from Fig. 9 in the Northern Hemisphere for the 30° W latitudinal cross section in particular, where relatively large apparent model disagreement at 350 hPa can be attributed to a slight downward shift in CMAM relative to EMAC, consistent with that found in Sects. 3 and 4. Relative to CMAM, the largest increases in EMAC are shifted equatorward ($\sim$ 10–40° N) and are collocated more closely with the region influenced by the subtropical jet stream (e.g. Manney and Hegglin, 2018), particularly in spring (MAM). In contrast, the largest changes in CMAM are generally poleward of 30° N, particularly at the 350 hPa level. The spatial distribution in the changes is also less zonally consistent than for EMAC, and this could reflect a greater influence in the eddy-driven (polar) jet stream in modulating such spatial variability.

Northern Hemisphere surface changes show greater regional variability due to the strong dependence of the surface environment as both a source of emission precursors and as a sink of ozone. In both seasons, the largest statistically significant increases can be found over south-east Asia (exceeding 6 ppbv locally), except for a small region of decrease over north-east China apparent only in CMAM. The 90° E latitudinal cross section in Fig. 9 intersects this region, showing the largest increase close to the southern flank of the Himalayas in each model during both MAM (+6– 10 ppbv) and SON (+4–6 ppbv), extending from the surface upwards towards the UTLS (350 hPa). A significant increase is also evident widely over oceanic regions, particularly in

CMAM and in SON where values exceed 2 ppbv. This could be attributable to a number of factors, including increases in emissions from international shipping, long-range transport from upstream precursor emission sources as well as enhanced subsidence in mid-latitudes due to the influence of subtropical high-pressure systems (e.g. the Azores High and the North Pacific High) which may have expanded and intensified in recent decades (Li et al., 2011, 2012). Long-range transport has a clear dominant influence over the Pacific sector, as expected due to the rapid advection from this region. Given recent emission controls in North America, and therefore smaller changes in surface ozone, this factor would be less influential over the Atlantic. Across Europe, there is a large discrepancy in the long-term changes between the two models, with negligible change in EMAC (or even slightly negative in MAM) but a considerable increase ($\sim$ 2–6 ppbv) in CMAM in both seasons. Figure 10 later shows that this difference is at least partly related to the simulated downward flux of stratospheric ozone in each model during spring (MAM) but not in autumn (SON), with the remaining difference likely related to the chemistry schemes in each model. It is, however, noted from Jöckel et al. (2016) that the timing of road traffic emissions is offset in this EMAC simulation, leading to a slight underestimation of tropospheric partial-column ozone (up to $\sim$ 1.5 DU in Northern Hemisphere mid-latitudes during boreal summer between 2000 and 2013), but any impact on calculated ozone changes or trends has not yet been quantified.

Smaller changes are typically found over the tropics and across parts of the Southern Hemisphere in both models (Fig. 8), but particularly in CMAM and during autumn (MAM) when changes are near-zero or even negative. Between 0° N and 30° S, a continuous region of statistically significant increase in ozone ($\sim$ 2–6 ppbv) is, however, apparent along a north-west to south-east axis over the Pacific, South America and South Atlantic at both 350 and 500 hPa, which is largest and most coherent in EMAC and during SON, particularly over the Pacific Ocean. The geographical orientation of this feature is consistent with the climatological positioning of the Southern Hemisphere subtropical jet stream. Over Africa, a relatively small region of decrease (along or slightly south of the Equator) is present in both seasons, in both models at 350 and 500 hPa. The largest decreases are evident in SON, where locally ozone has decreased at a rate of 4–6 ppbv. This feature is not always statistically significant, likely due to its small-scale and subsequent enhanced sensitivity to interannual variability. The latitudinal cross section through 30° E in Fig. 9 shows this feature to be most pronounced in the mid- to upper troposphere in each model (even absent in CMAM in the lower troposphere). The bimodal structure of the changes in ozone (with an increase to the south of this region) is again consistent with a poleward shift in the subtropical upper-tropospheric jets as found by Manney and Hegglin (2018) and the location of STE. During autumn (MAM), CMAM shows a de-

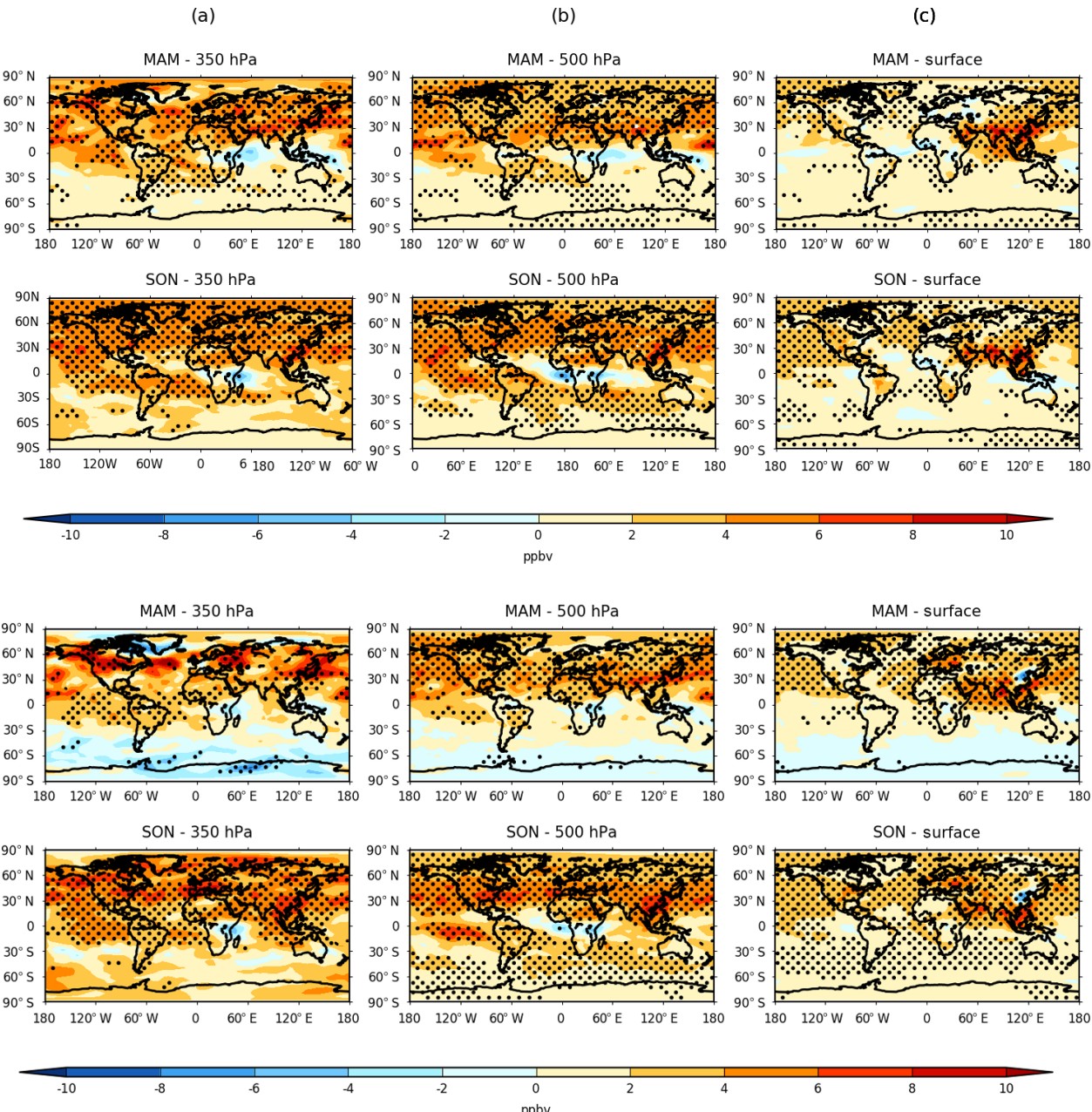

**Figure 8.** Seasonal change in EMAC (top) and CMAM (bottom) ozone ($O_3$) VMR (ppbv) between 1980–1989 and 2001–2010 for MAM and SON at **(a)** 350 hPa, **(b)** 500 hPa and **(c)** the surface model level. Stippling denotes regions of statistical significance according to a paired two-sided $t$ test ($p < 0.05$).

crease over much of the extratropics (statistically significant in places at 350 hPa) which could be related to the effects of stratospheric ozone depletion and the influence this may have on STE of ozone. Ozone depletion principally occurs, however, during spring (SON), so any apparent delayed impact on tropospheric ozone would need to be investigated further. Indeed, both models (but particularly CMAM) show widespread, statistically significant increases across much of the Southern Hemisphere extratropics during this season at 500 hPa and to a lesser extent at the surface, which appears related to the larger, regional increases in the subtropics, likely through long-range transport and entrainment around the hemisphere by upper-level winds. The relatively insignificant changes at 350 hPa and changes in $O_3S$ (Sect. 5.2) imply that this increase is tropospheric driven.

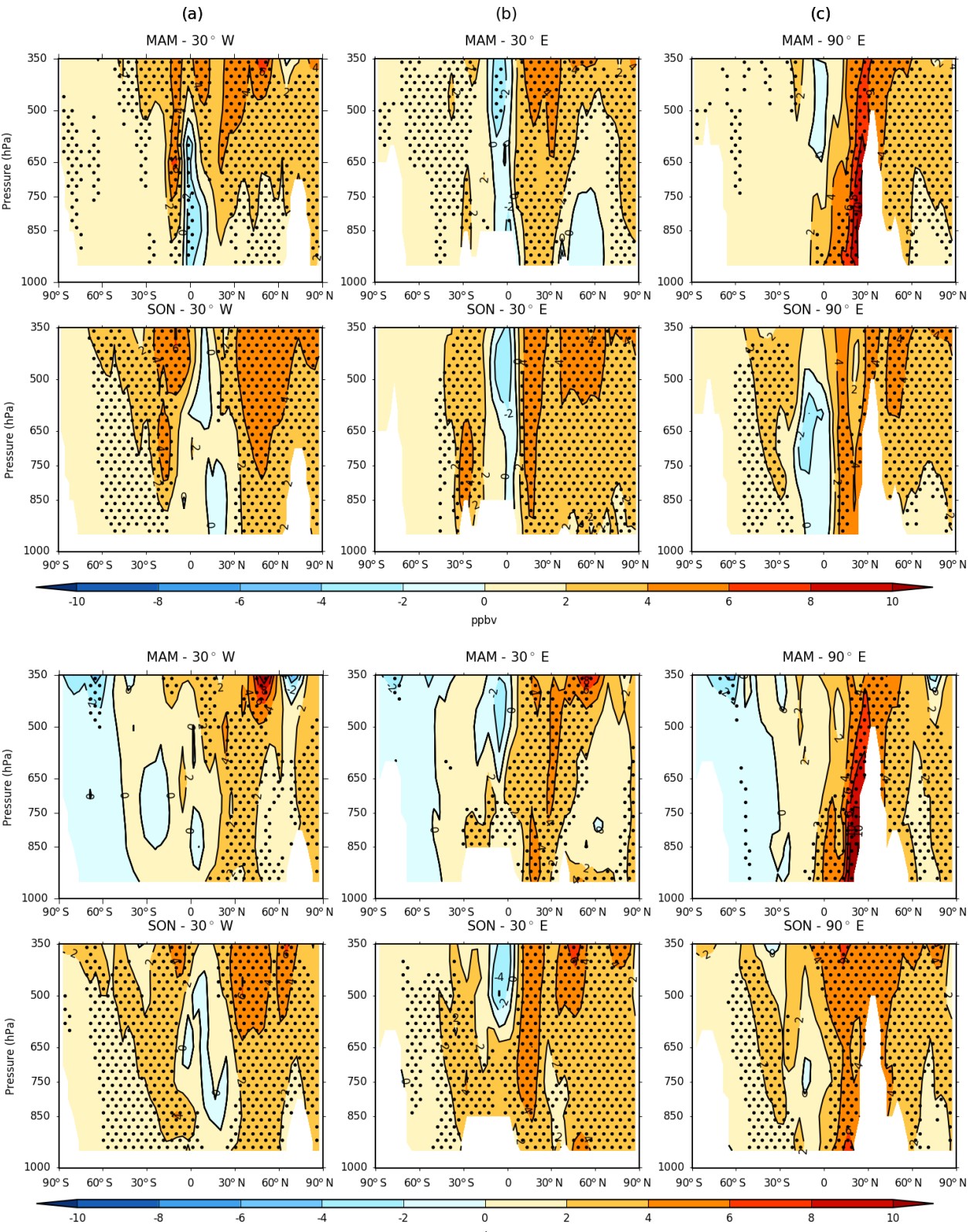

**Figure 9.** Longitudinal cross sections of the seasonal change in the vertical distribution of ozone ($O_3$) VMR (ppbv) from EMAC (top) and CMAM (bottom) between 1980–1989 and 2001–2010 for MAM and SON at **(a)** 30° W, **(b)** 30° E and **(c)** 90° E. Stippling denotes regions of statistical significance according to a paired two-sided $t$ test ($p < 0.05$).

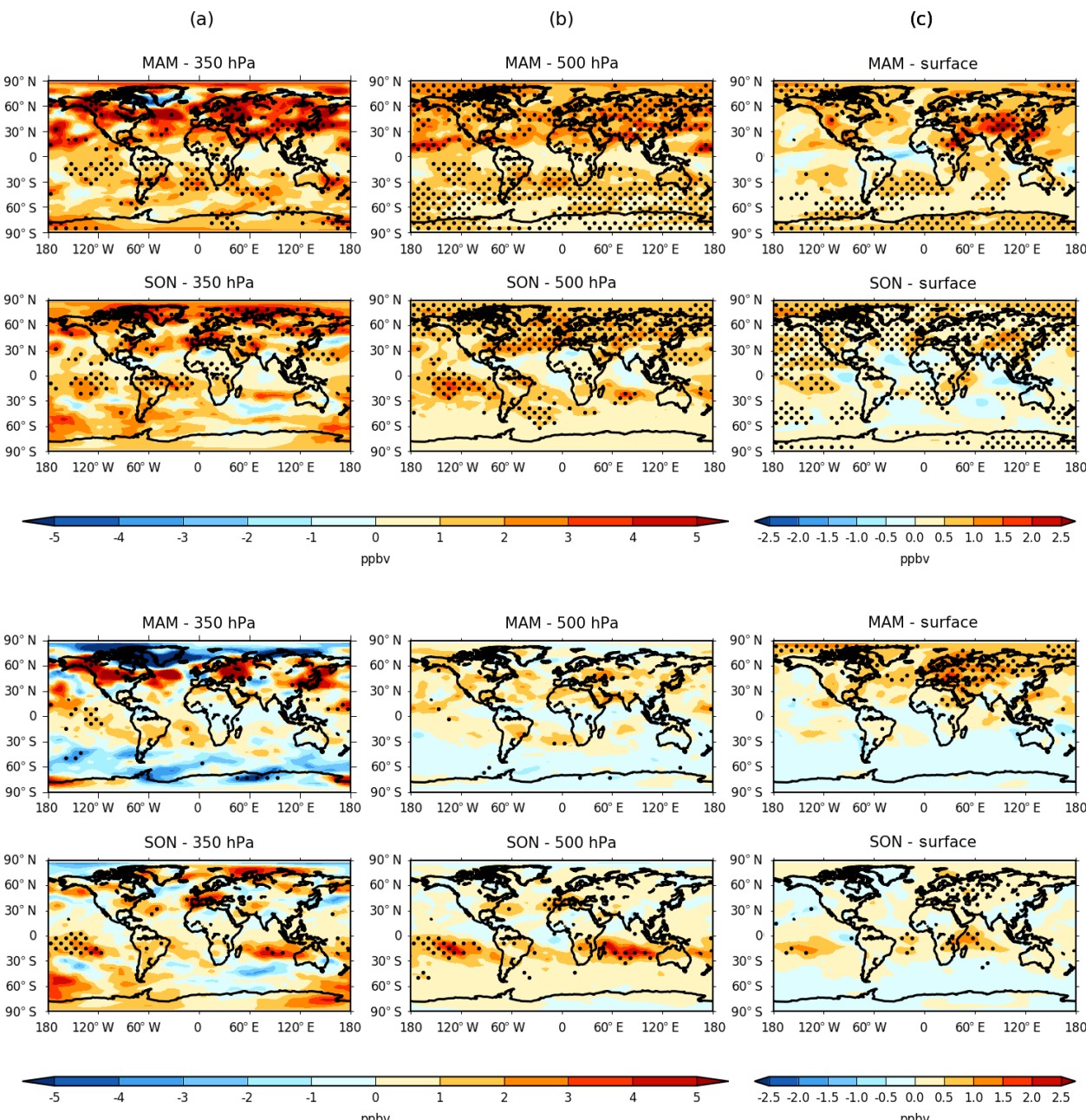

**Figure 10.** Seasonal change in EMAC (top) and CMAM (bottom) stratospheric ozone (O$_3$S) VMR (ppbv) between 1980–1989 and 2001–2010 for MAM and SON at **(a)** 350 hPa, **(b)** 500 hPa and **(c)** the surface model level. Stippling denotes regions of statistical significance according to a paired two-sided $t$ test ($p < 0.05$). Note the scale difference between **(a–b)** and **(c)**.

## 5.2 O$_3$S change (1980–1989 to 2001–2010)

The long-term changes in the corresponding stratospheric ozone (O$_3$S) model tracers shown in Figs. 10 and 11 for MAM and SON (and Fig. S8 and S9 for DJF and JJA) help attribute the long-term changes in O$_3$ described above primarily either to changes in STE or to changes occurring in the troposphere, such as the photochemical production of ozone from precursors as well as changing tropospheric

transport regimes. Similarly to the changes in O$_3$, both the largest spatial variability and changes in O$_3$S are evident towards the upper troposphere (350 hPa), particularly in the Northern Hemisphere where an overall increase can again be seen between both periods. The largest increases in O$_3$S span the mid-latitudes in the Northern Hemisphere (particularly during MAM), with extensive regions of +3–5 ppbv or greater and +2–4 ppbv in both models during spring (MAM) and autumn (SON) respectively, although statistical signifi-

cance is often lacking in CMAM especially, indicating the high level of inter-annual variability in upper-tropospheric dynamics. This can again be inferred from the spatial change patterns in the upper troposphere in the latitudinal cross sections in Fig. 11 (Fig. S9) but most notably along the $30°$ W meridian, where subtle shifts in the height of the tropopause, tropopause pressures of up to 30–50 hPa higher in CMAM (Hegglin et al., 2010) and associated sharp gradients in ozone VMR may at least partly explain the large discrepancies between the models in both the sign and magnitude of changes for any given region at the 350 hPa pressure level. Both models are, however, consistent in showing statistically significant increases in the regions of the subtropical jet, but particularly in EMAC, which is also evident in the mid-troposphere (500 hPa). In contrast, the models differ significantly at high latitudes, especially in MAM when CMAM shows a large decrease ($> -5$ ppbv) over parts of NE Canada, Southern Greenland and Northern Siberia.

Although EMAC shows a few localised regions of slight decrease, which are spatially collocated with CMAM, the model is dominated by an increase in $O_3S$ at these latitudes. Together with inter-model discrepancies in tropopause height, the spatial distribution in changes during MAM (most notably in CMAM) could reflect an equatorward shift in the mean position of the eddy-driven polar jet stream over time and the subsequent area of preferential downward STE, which has been identified through trend analyses using reanalysis datasets (Manney and Hegglin, 2018). Indeed, an equatorward trend of $\sim -0.4°$ dec$^{-1}$ in the jet latitude has also been calculated for both models for the period 1960–2000 in a recent study by Son et al. (2018), as determined by the maxima in the 850 hPa zonal mean zonal wind, although this trend is typically poleward for most other CCMI models. Conversely, changes at 500 hPa are much more spatially uniform, although large differences remain between the two models. Surface changes in $O_3S$ on the other hand are generally modest, with the large role of precursor emissions in contributing to the increase in $O_3$ (Fig. 8) obvious across many regions, most notably over SE Asia, when comparing such changes with the calculated changes in the model $O_3S$ tracers. Nonetheless, some regions (e.g. western North America and Eurasia) show an increase of 1–2 ppbv in MAM (locally significant), which represents a large fraction of the corresponding increase in $O_3$ (or even an offset of a slight negative change over parts of Europe in EMAC) as previously shown in Fig. 8. The main difference between the two models is the larger relative increase in $O_3S$ in EMAC across much of the Middle East and central southern Asia and conversely across much of Europe and western Eurasia in CMAM. The former difference is additionally highlighted in the $90°$ E transect (Fig. 11) which intersects the Himalayan region, although both models show a statistically significant increase ($> 1$ ppbv) in spring (MAM) along the northward flank of the mountain range which represents a minimum contribution of $\sim 25\%–30\%$ to the surface ozone change of

2–4 ppbv (Fig. 9). Regional discrepancies are smaller in SON with a general, albeit smaller, increase in $O_3S$ ($\sim +0$–1 ppbv) apparent, which is most pronounced in EMAC.

Changes in $O_3S$ across the tropics at both 350 and 500 hPa are generally small, consistent and of similar magnitude between each model during both MAM and SON, reflecting the absence of influence from the stratosphere (typical tropical tropopause altitude of $\sim 100$ hPa in the tropics) and a general upwelling regime. In the Southern Hemisphere subtropics, however, both models show hemispheric-wide, sometimes statistically significant increases in $O_3S$ on the order of $\sim +1$–4 ppbv centred between 10 and $30°$ S, except in CMAM during MAM when any increase is confined over South America and adjacent oceanic regions. Such a zonal structure in the spatial trend patterns is strongly supportive of influence from the subtropical jet stream, with the largest changes offset slightly equatorward of the climatological mean position in both seasons as identified in the literature (Langford, 1999; Manney and Hegglin, 2018). Indeed, preferential transport from the stratosphere to the troposphere has a known tendency to occur on the equatorward side of the jet (Lamarque and Hess, 2003). The calculated changes in the $O_3S$ tracer confirm that the $O_3$ changes (Fig. 8) are primarily driven ($> 50\%$) by an enhanced influence from the stratosphere, with the increase largest in CMAM during austral spring (SON) in likely association with an increased lower branch in the BDC in this model, which is more pronounced in the Southern Hemisphere (Hegglin et al., 2014; Haenel et al., 2015). Poleward of $30°$ S, changes are weak and generally insignificant at 500 hPa, with CMAM exhibiting an overall slight decrease during MAM and also in SON over Antarctica, whilst EMAC displays a slight increase generally (only exceeding 1 ppbv on a local basis), most pronounced in MAM where changes are significant in places. The spatial change patterns are broadly similar at 350 hPa, although spatial variability is considerably larger and complex patterns emerge, with particularly large discrepancies during MAM between each model. The differential spatial change patterns in each model at this height could be attributable to a range of factors such as the simulation of stratospheric ozone depletion, changes in the BDC between the two time periods and differences in tropopause altitude in each model. Surface changes in $O_3S$ across the Southern Hemisphere are small (and insignificant in places), although two localised regions of statistically significant increase (locally $> 1$ ppbv in CMAM) emerge in SON in the tropics, in the central South Pacific and over part of the western Indian Ocean and eastern Africa. The latter region is captured in the $30°$ E latitudinal cross section (Fig. 11) in CMAM especially, with a clear downward pathway in evidence coupling changes in $O_3S$ from the tropopause to the surface. Both regions are collocated spatially with the area of largest increase in $O_3S$ at both 350 and 500 hPa in the Southern Hemisphere, indicating that the influence of enhanced STE of ozone during SON between 1980–1989 and 2001–2010 is able to penetrate deep

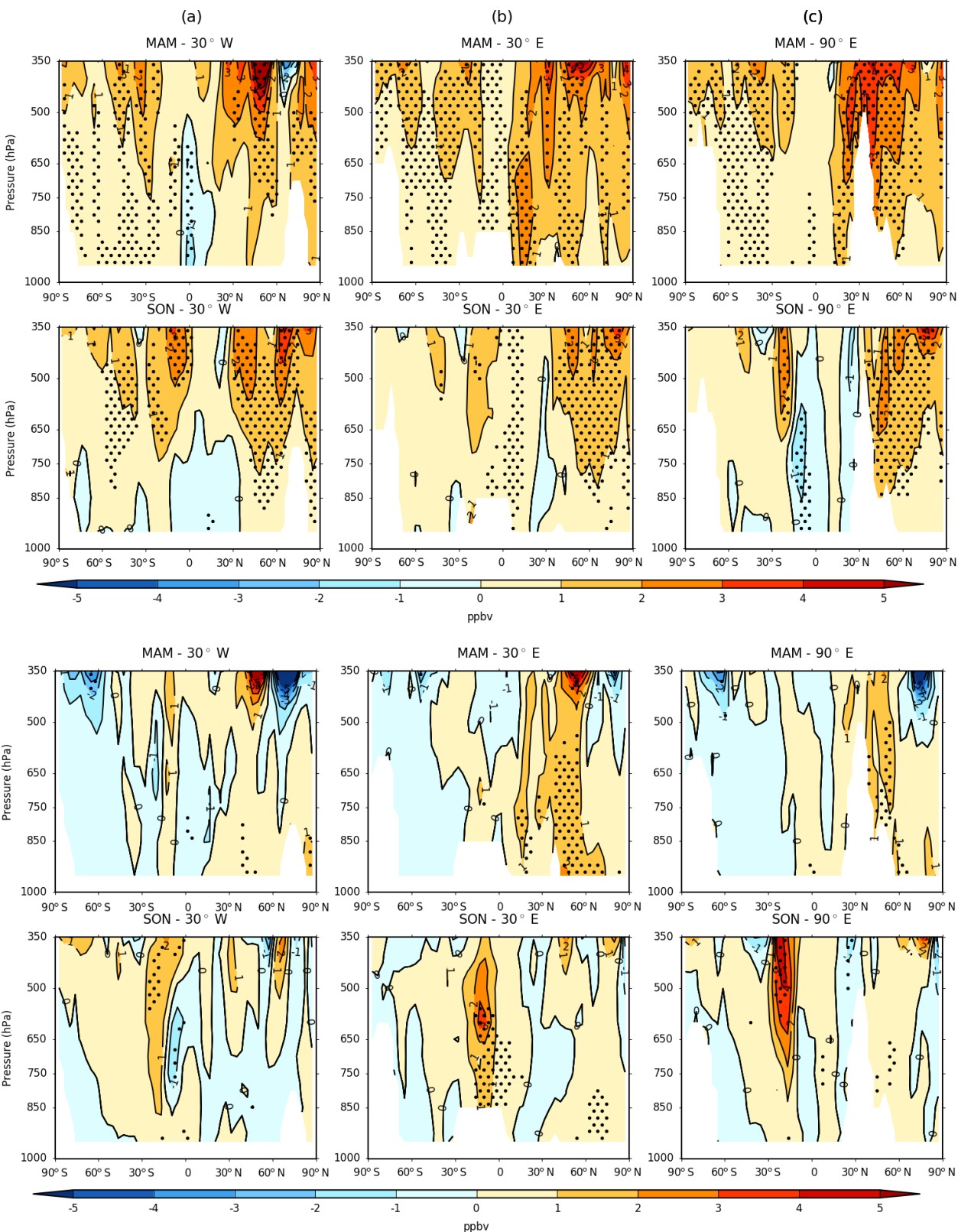

**Figure 11.** Longitudinal cross sections of the seasonal change in the vertical distribution of stratospheric ozone ($O_3S$) VMR (ppbv) from EMAC (top) and CMAM (bottom) between 1980–1989 and 2001–2010 for MAM and SON at **(a)** 30° W, **(b)** 30° E and **(c)** 90° E. Stippling denotes regions of statistical significance according to a paired two-sided $t$ test ($p < 0.05$).

into the PBL in these regions, explaining most of the increase in the model $O_3$ tracers locally here.

## 5.3 Summary

To summarise, changes in seasonal mean tropospheric ozone are generally positive between 1980–1989 and 2001–2010 in both models, with a maximum increase of $\sim 5\%$–$10\%$ corresponding to approximately 4–6 ppbv over the Northern Hemisphere and 2–6 ppbv over the Southern Hemisphere subtropics during springtime in both the middle (500 hPa) and upper troposphere (350 hPa). A significant stratospheric contribution to such increase is found here of up to 3–5 (1–4) ppbv during this season ($\sim 50\%$–$80\%$), although significant inter-model disagreement exists in the magnitude and sometimes the sign of the attributable change in ozone due to the stratosphere for any given region or season. This is particularly the case in the extratropics, where different responses to transport likely arise in each model resulting from nudging to specified dynamics as captured in ERA-Interim. Both the ozone ($O_3$) and stratospheric ozone ($O_3S$) tracers exhibit a preferential increase in the subtropics in EMAC and extratropics in CMAM, which may reflect the relative importance of the subtropical and polar jet streams respectively. This difference is, however, larger in the former case, which implies that the higher amounts of simulated ozone from precursor emissions in EMAC, particularly in the Northern Hemisphere subtropics, propagate upward from the surface and longitudinally due to the influence of these two jet streams, contributing to this difference. In the tropics and Southern Hemisphere extratropics, on the other hand, estimated changes are typically small and insignificant, with some indication of a decrease over high latitudes in CMAM. This could be attributable to the influence of stratospheric ozone depletion, but this requires further investigation given the lack of model agreement and the largest decrease in autumn (MAM), which is not consistent with the timing of the springtime stratospheric ozone hole. Although surface ozone changes are dominated by regional changes in precursor emissions between the two periods – the largest, statistically significant increases ($> 6$ ppbv) being over south-east Asia – the changing influence from the stratosphere is also shown to be highly significant. Indeed, the global area of statistical significance in the calculated $O_3S$ changes typically increases from the upper troposphere (350 hPa) to the surface. Increases in surface ozone driven by the stratosphere are estimated to be up to 1–2 ppbv between the two periods in the Northern Hemisphere, although this is highly variable both regionally and seasonally and between each model. In relative terms, the stratosphere can be seen to typically explain $\sim 25\%$–$30\%$ of the surface change over some regions such as the Himalayas, although locally it may represent the dominant driver ($> 50\%$) where changes in emission precursors are negligible or even declining due to the enforcement of air quality regulations over regions such

as western Europe and eastern North America. The stratospheric influence over changes in tropospheric ozone could be overestimated in the case of CMAM, which has deficiencies in the representation of tropospheric chemistry, although both models contain a well-resolved stratosphere and in the case of EMAC, a comprehensive tropospheric chemistry scheme. It is claimed by Neu et al. (2014) that models without comprehensive tropospheric chemistry cannot be used to estimate stratospheric influence, since a much larger response to tropospheric ozone is found in such models, although we find that EMAC shows a larger increase in stratospherically tagged ozone ($O_3S$) which challenges this statement. The much smaller STE response found in their study, which shows only a modest 2 % change in Northern Hemisphere mid-latitude tropospheric ozone to a $\sim 40\%$ variation in the strength of the stratospheric circulation, is also inferred from variability that occurs on inter-annual timescales due to ENSO and the QBO, which is used a proxy for the mean change in the stratospheric circulation this century. Therefore, the calculated changes presented here would also question the assumption that inter-annual variations in ENSO and the QBO constitute a representative surrogate for long-term changes anticipated due to climate change.

## 6 Conclusions

Seasonal variability, stratospheric influence and recent changes in tropospheric ozone are evaluated in this study using two state-of-the-art CCMs, which have the added provision of tagged stratospheric ozone tracer simulations. This study finds evidence that both CCMs are broadly consistent and agree with satellite (OMI) observations and limited in situ (ozone-sonde) profile measurements over the 2005–2010 common baseline period in simulating both the geographical variability and seasonality in tropospheric sub-column (1000–450 hPa) ozone. Inherent, systematic biases (with a strong seasonal dependence) are, however, shown to exist in each model. EMAC is characterised by an overall positive bias with respect to OMI, largest in Northern Hemisphere low latitudes during springtime ($\sim +2$–$8$ DU or $+10\%$–$30\%$). In contrast, CMAM shows no obvious overall bias ($\sim -4$ to $+4$ DU or $-20\%$ to $+20\%$) but has significant regional, latitudinal and seasonal variability in both the sign and magnitude of the bias relative to OMI. In CMAM, the mid-latitude seasonal evolution of the biases relative to OMI (Fig. 2) shows larger consistency prior to the application of the satellite (OMI) AKs, with respect to ozone-sondes for three different extratropical regions (Fig. 4), which is contrary to that expected through accounting for the observation geometry of the satellite. Whilst the application of AKs serves to slightly mitigate the positive tropospheric bias in mid-latitudes in EMAC, the negative bias in CMAM is generally converted to a positive bias in mid- to high latitudes. Comparisons with ozone-sondes indicate that the low tro-

pospheric bias in CMAM, likely related to the simplicity of the model chemistry scheme, is offset due to an inherent high ozone bias in the lowermost stratosphere (as high as 40 %–60 %). This leads to excessive downward smearing of ozone into the troposphere as a result of applying satellite (OMI) AKs, necessary to compare both model simulation and OMI satellite measurements equivalently. This highlights an important trade-off in the application of satellite AKs for model–measurement comparison analyses of tropospheric ozone where biases in lower-stratospheric ozone are known to exist. This evaluation implies that in certain circumstances, the application of AKs would not be advocated where model biases in lower-stratospheric ozone are sufficiently large due to anomalous vertical smearing. However, such a detailed quantitative evaluation would be needed to identify such cases. The high bias in mid-latitudes in EMAC could be explained by an overestimation of emissions in MACCity (a CMIP5-based inventory) (Hoesly et al., 2018), which although used in both models, leads to a higher bias in EMAC due to the comparatively complex tropospheric chemistry scheme in this model. Given that the largest tropospheric biases are equatorward of the region influenced by vertical smearing from the lowermost stratosphere, the two influences are more independent in this model. The relative importance of these drivers is regionally and seasonally dependent but serves to yield an overall lower bias in CMAM compared with EMAC. The influence of applying AKs is typically to increase the sub-column amount of tropospheric ozone (1000–450 hPa) in the extratropics by ∼ 1– 5 DU and ∼ 2–8 DU in EMAC and CMAM respectively, depending on season, whereas a slight decrease (∼ 0–1 DU) is induced in the tropics in all seasons. An exception to this is over the Southern Hemisphere high latitudes, where the increase is significantly lower due to influence of the ozone hole, particularly in austral spring (SON) when any increase is negligible (0–1 DU). It is important to note that like models, satellite retrieval platforms such as OMI have their own limitations, such as the susceptibility to instrument noise or retrieval errors (Levelt et al., 2006, 2018; Mielonen et al., 2015; Schenkeveld et al., 2017). It is suspected that this limitation is the cause of the large discrepancy in the seasonal composites of RSD, as a metric for the inter-annual variability, between OMI and the models, the latter of which is in closer agreement with that derived from ozone-sondes. A general consensus on the inter-annual variability in tropical tropospheric ozone is, however, found, with RSD values of over 10 % in some regions and seasons, consistent with the known influence of different teleconnections, most notably the QBO, which is estimated to influence tropical tropospheric ozone anomalies by as much as 10 %–20 % (8 ppbv) (Lee et al., 2010). Inconsistencies in a number of the model–OMI and model–ozone-sonde differences are also suspected to undermine the issue of resolution (in the case of models) and signal-to-noise ratio (in the case of OMI) in adequately resolving mesoscale features, such as local-scale pol-

lution plumes or stratospheric intrusion (tropopause folding) events, although this would be an area warranting further investigation.

Taking the above information (from the model–measurement comparison in Sect. 3) into account, the relatively long temporal span of the specified-dynamics CCM simulations was utilised to investigate the climatological stratospheric influence on tropospheric ozone and calculate estimated recent changes between 1980–1989 and 2001–2010. A clear difference in the strength and dominance of the shallow branch of the BDC is implied in each model, due to the large discrepancy in the burden of ozone in the extratropical lowermost stratosphere (∼ 50 %–100 % more ozone in CMAM compared with EMAC). The characterised biases with respect to ozone-sondes indicate that CMAM has a faster, shallower BDC compared to actuality, which can be inferred from the large lower-stratospheric ozone bias (∼ +20 %–60 %), whereas EMAC provides a more realistic simulation of the BDC, albeit perhaps one that is too conservative given a general negative ozone bias (up to 10 %–20 %) in the lower stratosphere. The difference in BDC simulation has implications for the simulated STE flux of ozone, with preferential downward transport in the subtropics in CMAM compared with the mid-latitudes in EMAC, particularly in the Southern Hemisphere subtropics and during springtime when the difference is as much as 10 %–25 % from the lower to upper troposphere. Compared to the model results of Lamarque et al. (1999), the CCM simulations examined here are in much closer agreement with ozone-sonde measurements, with biases no larger than 20 %, as evidenced on a zonally averaged, monthly basis in Fig. 7 (Fig. S5). This contrasts with a systematic underestimation of tropospheric ozone VMR by as much as 20 %–50 % in the CTM analysed in their study. Despite a significant fall in the correspondence between the seasonal evolution of the simulated ozone and stratospheric ozone component in the CCMs from the upper to the lower troposphere, the results show a significant stratospheric influence on even lower-tropospheric ozone – greater than 50 % in the wintertime extratropics, which contrasts with a modest 10 %–20 % estimated from the CTM in Lamarque et al. (1999).

Both models show an overall, statistically significant increase in ozone between 1980–1989 and 2001–2010, on the order of ∼ 5 %–10 %, or some 4–6 ppbv over the Northern Hemisphere and 2–6 ppbv over the Southern Hemisphere subtropics, in the middle to upper troposphere, with a preferential increase over the subtropics in EMAC compared to the extratropics in CMAM (most pronounced in spring). As estimated using stratospherically tagged ozone tracers from each model, the stratosphere is found to provide a substantial contribution ranging between 1 and 3 ppbv (∼ 20 %–50 %) in the mid-troposphere (500 hPa) and over 5 ppbv (∼ 50 %– 80 %) in the upper troposphere (350 hPa) across the Northern Hemisphere mid-latitudes, with a typical increase of 1–

4 ppbv ($\sim 50\,\%$–$80\,\%$) over the Southern Hemisphere sub-tropics at both pressure levels. Significant model disagreement, however, exists, particularly in the extratropical upper troposphere, likely due to known discrepancies in tropopause height (Hegglin et al., 2010) and the variability in upper-level dynamics, which may be further affected by the nudging applied to the models. Estimated changes in ozone and the stratospheric contribution, on the other hand, are generally small and insignificant in both equatorial and Southern Hemisphere extratropical regions. The spatial pattern of changes in surface ozone in contrast show a very different character, with the largest statistically significant increases over much of south-east Asia ($> 6$ ppbv) and a general increase of up to 2 ppbv or higher quite widely over Northern Hemisphere oceanic regions, but only very small, non-significant changes across the Southern Hemisphere. The influence from the stratosphere at the surface is seen to have a strong regional and seasonal dependence but is estimated to be as much as 1–2 ppbv during spring, which was estimated to be as large as $\sim 25\,\%$–$30\,\%$ along the northern flank of the Himalayan mountain range and greater than $50\,\%$ over a localised, relatively unpolluted region of eastern Africa and the western Indian Ocean. The situation is complicated in some regions, however, where near-zero or slight negative changes in ozone VMR are apparent in places such as western Europe and eastern North America, corresponding to an observed hiatus or slight fall in precursor emissions.

This study highlights some of the shortcomings of both the EMAC and CMAM CCMs as part of the IGAC–SPARC CCMI activity, as validated with respect to satellite observations from OMI and in situ ozone-sonde measurements, in simulating tropospheric ozone. In particular, the importance of a well-resolved stratosphere is clear in attaining a high level of model–measurement agreement and in terms of adequately representing stratospheric influence. For comparisons with satellite observational datasets, a well-resolved stratosphere is of paramount importance for the application of AKs, which smooth the vertical distribution of model-simulated ozone by smearing information down from the stratosphere to the troposphere. Using this derived knowledge, this study confirms the strong influence of the stratosphere in modulating tropospheric ozone and provides an indication that such influence may in fact be much larger than previously thought. Furthermore, recent changes in tropospheric ozone are shown to have a large attribution from the stratosphere, which is quantified here in relation to influence of changing precursor emissions. A general increase in the amount of stratospheric ozone in the troposphere between 1980–1989 and 2001–2010 according to both CCMs, which is statistically significant in some regions of the world such as western Eurasia, eastern North America, the South Pacific and the southern Indian Ocean, would be expected from observed long-term changes in the shallow branch of the BDC (Hegglin et al., 2014). Transit times have been found to exhibit a steady decrease, primarily due to accelerated transport within this branch of the residual circulation ($\sim 75\,\%$), with a smaller contribution from a shortening of the transit pathways ($\sim 25\,\%$) (Bönisch et al., 2011). Indeed, a strengthening of the BDC is postulated to be the main mechanism for an expected increase in STE under future climate change scenarios (Hegglin and Shepherd, 2009; Butchart et al., 2010), in addition to stratospheric ozone recovery (Zeng et al., 2010), which further highlights the need for an improved understanding of the relationship between STE and tropospheric ozone and accurate quantitative estimates. These findings thus have important implications for the enforcement of both current and future air quality regulations as well as in constraining estimates of tropospheric ozone radiative forcing.

*Data availability.* All CCM simulations analysed here are publicly available from CEDA/BADC via the CCMI data archive (http://www.ceda.ac.uk, Hegglin and Lamarque, 2015). L3 data from OMI using the RAL profiling scheme can be provided on request by contacting barry.latter@stfc.ac.uk. Ozone-sonde profile data are publicly accessible from the WOUDC database (http://www.woudc.org, WMO/GAW Ozone Monitoring Community, 2015).

*Supplement.* The supplement related to this article is available online at: https://doi.org/10.5194/acp-19-1-2019-supplement. TS9

*Author contributions.* RSW (the lead author) designed the research study, undertook the analyses and prepared the manuscript under the close supervision of MIH, with some additional supervisory support from BK. BL produced and provided access to OMI–RAL L3 data, with both BK and BL able to offer technical support on the satellite (OMI) dataset. Similarly, PJ and DAP provided technical support in relation to the EMAC and CMAM CCM simulation datasets respectively. All co-authors provided comments and suggestions helping RSW to greatly improve the quality of the manuscript.

*Competing interests.* The authors declare that they have no conflict of interest.

*Special issue statement.* This article is part of the special issue "Chemistry–Climate Modelling Initiative (CCMI) (ACP/AMT/ESSD/GMD inter-journal SI)". It is not associated with a conference.

*Acknowledgements.* Ryan S. Williams would like to personally thank the Natural Environmental Research Council (NERC) for funding the research and each co-author in providing their support and expertise in shaping the research study and the design of the manuscript. We acknowledge the modelling groups for making their simulations available for this analysis, the joint WCRP SPARC/IGAC Chemistry-Climate Model Initiative (CCMI) for

organising and coordinating the model data analysis activity, and the British Atmospheric Data Centre (BADC) for collecting and archiving the CCMI model output. The EMAC simulations were performed at the German Climate Computing Centre (DKRZ) through support from the Bundesministerium für Bildung und Forschung (BMBF). DKRZ and its scientific steering committee are gratefully acknowledged for providing the HPC and data archiving resources for the consortium project ESCiMo (Earth System Chemistry integrated Modelling).

Edited by: Tim Butler
Reviewed by: three anonymous referees

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

**Remarks from the language copy-editor**

CE1    Thanks for your clarification. If this doesn't abbreviate a fuller name, it's fine as it stands. Would you like us to include the supplement link you sent as a proper reference?

**Remarks from the typesetter**

TS1    This change needs to be approved by the editor.
TS2    This change needs to be approved by the editor.
TS3    This change needs to be approved by the editor.
TS4    This change needs to be approved by the editor.
TS5    This change needs to be approved by the editor.
TS6    This change needs to be approved by the editor.
TS7    This change needs to be approved by the editor.
TS8    This change needs to be approved by the editor.
TS9    Please note that the supplement link is a placeholder link during proofreading and will be updated upon publication.