# Peer review of "Characterising the seasonal and geographical variability in tropospheric ozone, stratospheric influence and recent changes"

_Atmospheric Chemistry and Physics, 2018_

## Referee Comment (RC1) · Anonymous Referee #1 · 4 Dec 2018

The paper by Williams et al. "Characterising the Seasonal and Geographical Variability of Tropospheric Ozone, Stratospheric Influence and Recent Changes" utilises satellite and ozonesonde observations and two chemistry-climate models to investigate the stratospheric influence on tropospheric ozone. The authors conclude that the influence of stratospheric on tropospheric ozone is larger than previously thought. The authors also assessed the tropospheric ozone over the periods of 1980-89 and 2001-2010, and find an overall significant increase in tropospheric ozone, and attribute 25-30% changes at the surface and 50-80% in the upper troposphere to the stratosphere-troposphere exchange. The paper is well written, and the analysis is generally thorough, but some clarifications and improvements are needed before the paper can be

accepted for publication in ACP. Detailed comments are listed below.

General comments:

A major concern is that this study includes only two CCMI model results, which reduces the robustness of the finding, especially in that "the influence of STE in the tropospheric ozone is larger than previously thought". Furthermore, using only simulations constrained with prescribed dynamics might obscure the changes due to dynamical feedbacks, especially when assessing long-term changes in ozone. Therefore, I suggest that the authors tone down the conclusion mentioned above, and instead focus on the uncertainty in the contribution of STE to the tropospheric ozone. The limitation of using prescribed dynamics CCM simulations should also be noted and discussed. A wider usage of CCMI models would address the first comment. As a minimum, the authors should give a reason for using only the two chosen models.

Regarding previous studies, I doubt that the paper by Lamarque et al. (1999) is still a very relevant reference that the authors focus their comparisons on, given that the approach used in Lamarque et al. (1999) was very simplistic compared to what can be achieved using more recent state-of-the-art CCMs. Also, there are a few more studies that have investigated the impact of STE on tropospheric ozone which the authors failed to cite, for example, Lelieveld and Dentener (2000), Hess et al. (2013), etc.

Jos Lelieveld and Frank J. Dentener, What controls tropospheric ozone? JGR, 105, P3531-3551 2000.

P. G. Hess and R. Zbinden, Stratospheric impact on tropospheric ozone variability and trends: 1990–2009, ACP, 13, 649–674, 2013.

Therefore, a more thorough review of the recent literature would be desirable.

Specific comments:

P2, L13: add "large" before "number"

P4, L17: please add references here

P5, L5-L20: Can you describe the models' characteristics in a more objective way here? Why do you choose these two models specifically (there are quite a few other models from CCMI that you could include)? Also describe the main differences between these two models.

P5, L28: Pease provide more details in chemical schemes used in EMAC.

P6, L4 & L21: Please provide more detailed information on how the O3S tracer is defined in terms of its chemical and dynamical nature in both models.

P7, L19-L21: Please clarify if the AKs have or have not been applied to the modelled and ozonesode data when you compare these two. It only makes sense to apply AKs when compare model/sonde data to the satellite data.

P7, L27-L29: I don't understand why "The 1000-450 hPa (0-5.5 km) OMI subcolumn data is considered a representative approximation of the full tropospheric ozone column amount, due to vertical smearing of information from above 450 hPa ($\sim$ 5.5 km)." is the case. Is it possible to show AKs?

P9, L1-L4: Please add references here. There are a series of publications on JOSIE by Smit et al.

Smit, H. G. J., and D. Kley (1998), JOSIE: The 1996 WMO International intercomparison of ozonesondes under quasi flight conditions in the environmental simulation chamber at JuÂĺlich, WMO Global Atmosphere Watch report series, No. 130 (Technical Document No. 926). World Meteorological Organization, Geneva.

Smit, H. G. J., and W. Straeter (2004a), JOSIE-1998, Performance of ECC Ozone Sondes of SPC-6A and ENSCI-Z Type, WMO Global Atmosphere Watch report series, No. 157 (Technical Document No. 1218), World Meteorological Organization, Geneva.

Smit, H. G. J., and W. Straeter (2004b), JOSIE-2000, JuÂĺlich Ozone Sonde Intercomparison Experiment 2000, The 2000 WMO international intercomparison of operating procedures for ECC-ozonesondes at the environmental simulation facility at JuÂÍlich, WMO Global Atmosphere Watch report series, No. 158 (Technical Document No. 1225), World Meteorological Organization, Geneva.

P10, L19: Ozone is not at its minimum in SON in the SH, but maximum. Biomass burning emissions and STE usually dominate the seasonality of SH O3.

P11, L21-23: can you provide more details on the difference between these two NOx emission datasets?

P12, L1-L2: This seems slightly mis-leading on the function of the AKs. The purpose of applying the AVKs is to compare like with like.

P12, L4-L7: Similar features seem exist in both models; it seems more likely due to transport barrier than STE (which the STE maximises in winter).

P12, L7-L9: Does the difference in chemical schemes between the two models play a role here?

P13, Fig 3: It is impossible to discern the RSD of ozonesonde data denoted as circles, due to a uniformed colour scale.

P14, Fig 4: The value of 100 ppbv O3 seems a bit too low for defining the tropopause. Using 100 ppbv O3 also deviates from the definition by Bethan et al. (1996) (cited in caption), which is based on the ozone gradient, defined as the minimum altitude where the vertical gradient of the O3 VMR is greater than 60 ppb/km, remains so for a further 200 m, and the O3 mixing ratio is greater than 80 ppbv, exceeding 110 ppbv immediately above the tropopause.

P15, L16-L32: Please note the figures that you are discussing throughout this paragraph. Also, the description/discussion in this paragraph can be simplified to focus on the main points.

P16, L1-L2: again, please can you refer to the figure(s) that you are discussing.

P16, L27: Do you also apply AVKs to model data when compare them with ozonesonde data? If so, it is not necessary.

P16, L28: do you mean "simplified" tropospheric chemistry scheme? It is unusual to use the word "conservative".

P16, L29-L31: which comparison/figures you are talking about here? Please make it clear by referring to figures.

P16, L32-L33: It seems lacking context regarding "since vertical smearing of information is far more limited due to a higher tropopause.". Could you be specific? Where is the information regarding a higher tropopause?

P17: L1: "must induce" should be "must have induced"

P17: L3-L7: Showing the AKs might help with the discussion here.

P21, L31: Please provide details on how you map the model data to ozonesonde measurements shown in Fig 7?

P23, L1: it is too general to say that ". . . are evident in the contemporary CCM simulation" while only two CCMs are used here.

P24, L11: What do you mean "even lower tropospheric ozone"?

P24, L20: What is the rationale for choosing these longitudes?

P25, Fig 8: There are large areas in the SH that are denoted significant in CMAM (SON 500 hPa and SON surface plots), which are not reflected in the relevant discussion. Please check.

P27, L22: Please note which figure(s) you are discussing here? Is Fig 8?

P30, L10-L12: is "subtle shifts in the height of tropopause" shown anywhere?

P34, L10-L12: What are the reference variables for these percentage changes? P35, L2: Please specify re "some regions of the world".

---

## Referee Comment (RC2) · Anonymous Referee #3 · 18 Dec 2018

This is an interesting and useful analysis, but needs to be put in context and contrasted with other recent work before it is published. The authors would also be well-advised to better qualify the limitations of their study, in particular with regard to tropospheric chemistry.

Detailed remarks:

Page 3, lines 21-25: (and elsewhere) the authors focus on Lamarque et al. (1999), a 20-year old study. There are much more recent modeling studies that show larger net influence of STE. For example, Figure 6 of Banerjee et al. (2016) looks very much like Figure 5 of this manuscript. Doubtless similar plots for other models exist. If the authors

wish to make the case for their result "that the stratospheric influence on tropospheric ozone is larger than previously thought", they need to quote recent studies that show smaller influence.

Line 34: Similarly, the authors cite only one observational study, and for the Southern Ocean. There have been many such studies, e.g. Dibb et al. (1994); Elbern et al. (1997); Stohl et al. (2000); Zanis et al. (2003); Colette and Ancellet (2005); Cooper et al. (2006); Thompson et al. (2007a,b); Cristofanelli et al., (2010); Tarasick et al., (2018). Citing some of these would not only be appropriate, but would support the authors' point, as in general they find modest influence of STE on lower tropospheric ozone levels.

Page 4, lines 10-11: While I agree that there are certainly major limitations with the accuracy of retrieved tropospheric ozone from spaceborne instruments, I take issue with the statement "...scientists must instead rely on tools such as chemistry climate models (CCMs) to fill in the gaps in our understanding of the global distribution of tropospheric ozone". NO, NO, NO! Models are sophisticated data visualization tools: they contain (at best) all that we know about the atmosphere. They allow us insights and interpretation that would not be possible without them, but they do not contain anything that we don't, collectively, already know.

Page 8, lines 10-11: I thought the main problem was lack of photons that actually penetrate this far, as well as lack of contrast in the scattered spectrum, compared to a few km higher up.

Page 8, line 18: It seems odd to cite satellite papers for generic facts about the global ozonesonde network. Liu et al. (2013b) has a good discussion, with a map and table of sites. In line 26, the proper reference here is Smit et al. (2007), although the others are fine as additional references. On the next page (line 2), citations are required for the WMO & JOSIE campaigns.

Page 9: I find the statement in lines 28-30 quite remarkable. I believe there are many

studies showing that long-range transport of ozone and its precursors are the dominant source of ozone in remote areas.

Page 17 (top paragraph), and elsewhere: the authors put a lot of effort into explaining the effects of "vertical smearing". Of course they need to consider OMI AKs when comparing to OMI data, but perhaps they would find it easier to use a 3D ozonesonde-based dataset, like Liu et al. (2013a,b).

Page 22 (Figure 7a): How are these plots produced from ozonesonde data?

Page 23, line 22: The "significant difference in the strength and dominance of the shallow branch of the BDC in each model" needs more explanation. It is first mentioned here, in the Summary.

Page 31 (Summary): The authors should consider, and discuss, the differences between their results and the much smaller STE response found by Neu et al. (2014). In particular, Neu et al. claim that larger responses of tropospheric ozone to STE are found in models without comprehensive tropospheric chemistry.

Page 32, lines 31-34: Discussing the comparison before the AKs are applied makes little sense, and should be omitted.

Page 33, line 26: See previous comment, page 23, line 22.

Minor points:

Page 5, lines 11, 12: Typographical errors.

Page 5, line 34: The solar cycle evolves?

Page 6, line 9: Typographical error.

Page 6, line 19: "Compared with EMAC"?

Page 8, line 20: Actually the WOUDC also has data for Indian, Brewer-GDR, carbon-iodine and Regener sondes.

Page 8, line 33: Local air pollution (SO2) is not a significant source of error in recent decades, except in unusual circumstances (volcanoes).

Page 10, line 19: Maximum, not minimum.

Page 12, line 1: "An effect..." Not "The effect..."

Page 15, line 2: Minima is plural.

Figure 3 caption: "data" is plural.

Page 23, lines 28 & 31: Not clear which model is being discussed.

References

Banerjee A., A.C. Maycock, A.T. Archibald, N.L. Abraham, P. Telford, P. Braesicke, J.A. Pyle (2016), Drivers of changes in stratospheric and tropospheric ozone between year 2000 and 2100, Atmos. Chem. Phys., 16, 2727-2746, https://doi.org/10.5194/acp-16-2727-2016.

Cristofanelli, P., Bracci, A., Sprenger, M., Marinoni, A., Bonafè, U., Calzolari, F., Duchi, R., Laj, P., Pichon, J.M., Roccato, F., Venzac, H., Vuillermoz, E., and Bonasoni, P.: Tropospheric ozone variations at the Nepal Climate Observatory-Pyramid (Himalayas, 5079 m a.s.l.) and influence of deep stratospheric intrusion events, Atmos. Chem. Phys., 10, 6537-6549, doi:10.5194/acp-10-6537-2010, 2010.

Colette, A., and G. Ancellet (2005), Impact of vertical transport processes on the tropospheric ozone layering above Europe. Part II: Climatological analysis of the past 30 years, Atmos. Environ., 39(29), 5423–5435, doi:10.1016/j.atmosenv.2005.06.015.

Cooper, O.R, A. Stohl, M. Trainer, A. Thompson, J.C. Witte, S.J. Oltmans, G. Morris, K.E. Pickering, J.H. Crawford, Gao Chen, R.C. Cohen, T.H. Bertram, P. Wooldridge, A. Perring, W.H. Brune, J. Merrill, J. L. Moody, D. Tarasick, P. Nédélec, G. Forbes, M. J. Newchurch, F. J. Schmidlin, B. J. Johnson, S. Turquety, S. L. Baughcum, X. Ren, F. C. Fehsenfeld, J. F. Meagher, N. Spichtinger, C.C. Brown, S.A. McKeen,

I.S. McDermid, and T. Leblanc (2006), Large upper tropospheric ozone enhancements above mid-latitude North America during summer: In situ evidence from the IONS and MOZAIC ozone measurement network, J. Geophys. Res., 111, D24S05, doi:10.1029/2006JD007306.

Dibb, J.E., L.D. Meeker, R.C. Finkel, J.R. Southon, M.W. Caffee and L.A. Barrie (1994), Estimation of stratospheric input to the Arctic troposphere: 7Be and 10Be aerosols at Alert, Canada, J. Geophys. Res., 99, 12,855–12,864.

Elbern, H., J. Kowol, R. Sladkovic and A. Ebel (1997), Deep stratospheric intrusions: A statistical assessment with model guided analysis, Atmos. Environ., 31, No. 19, 3207-3226.

Liu, G., J.J. Liu, D.W. Tarasick, V.E. Fioletov, J.J. Jin, O. Moeni, X. Liu, C.E. Sioris and M. Osman (2013a), A global tropospheric ozone climatology from trajectory-mapped ozone soundings, Atmos. Chem. Phys. 13, 10659-10675, doi:10.5194/acp-13-10659-2013.

Liu, J., D.W. Tarasick, V.E. Fioletov, C. McLinden T. Zhao, S. Gong, C. Sioris, J. Jin, G. Liu, and O. Moeini (2013b), A Global Ozone Climatology from Ozone Soundings via Trajectory Mapping: A Stratospheric Perspective, Atmos. Chem. Phys., 13, 11441-11464, doi:10.5194/acp-13-11441-2013.

Neu, J.L., T. Flury, G.L. Manney, M.L. Santee, N.J. Livesey and J. Worden (2014), Tropospheric ozone variations governed by changes in the stratospheric circulation. Nature Geosci. 7, 340–344, doi: 10.1038/NGEO2138.

Smit, H.G.J., W. Straeter, B. Johnson, S. Oltmans, J. Davies, D.W. Tarasick, B. Hoegger, R. Stubi, F. Schmidlin, T. Northam, A. Thompson, J. Witte, I. Boyd and F. Posny (2007) Assessment of the performance of ECC-ozonesondes under quasi-flight conditions in the environmental simulation chamber: Insights from the Juelich Ozone Sonde Intercomparison Experiment (JOSIE), J. Geophys Res., 112, D19306,

doi:10.1029/2006JD007308.

Stohl, A., N. Spichtinger-Rakowsky, P. Bonasoni, H. Feldmann, M. Memmesheimer, H.E. Scheel, T. Trickl, S. Hubener, W. Ringer and M. Mandl (2000), The influence of stratospheric intrusions on alpine ozone concentrations, Atmos. Environ., 34, 1323–1354.

Tarasick, D.W., T.K. Carey-Smith, W.K. Hocking, O. Moeini, H. He, J. Liu, M. Osman, A.M. Thompson, B. Johnson, S.J. Oltmans and J.T. Merrill (2018), Quantifying stratosphere-troposphere transport of ozone using balloon-borne ozonesondes, radar windprofilers and trajectory models, Atmos. Environ., 198, 496-509, https://doi.org/10.1016/j.atmosenv.2018.10.040.

Thompson, A.M., J.B. Stone, J.C. Witte, R.B. Pierce , R.B. Chatfield, S.J Oltmans, O.R. Cooper, B.F. Taubman, B.J. Johnson, E. Joseph, T.L. Kucsera, J.T. Merrill, G. Morris, S. Hersey, M.J. Newchurch, F.J. Schmidlin, D.W. Tarasick, V. Thouret and J.-P. Cammas (2007a), Intercontinental Chemical Transport Experiment Ozonesonde Network Study (IONS) 2004: 1. Summertime upper troposphere/lower stratosphere ozone over northeastern North America, J. Geophys. Res., 112, D12S12, https://doi.org/10.1029/2006JD007441.

Thompson, A.M., J.B. Stone, J.C. Witte, S. Miller, S.J Oltmans, T.L. Kucsera, J.T. Merrill, G. Forbes, D.W. Tarasick, E. Joseph, F.J. Schmidlin, W.W. MacMillan, J. Warner, E. Hintsa and J. Johnson (2007b), Intercontinental Chemical Transport Experiment Ozonesonde Network Study (IONS) 2004: 2. Tropospheric ozone budgets and variability over northeastern North America„ J. Geophys. Res., 112, D12S13, doi:10.1029/2006JD007670.

Zanis P., T. Trickl, A. Stohl, H.Wernli, O. Cooper, C. Zerefos, H. Gaeggeler, C. Schnabel, L. Tobler, P. Kubik, A. Priller, H. E. Scheel, H.J. Kanter, P. Cristofanelli, C. Forster, P. James, E. Gerasopoulos, A. Delcloo, A. Papayannis and H. Claude, Forecast, observation and modelling of a deep stratospheric intrusion event over Europe, Atmos.

Chem. Phys., 3, 763-777, 2003.

---

## Referee Comment (RC3) · Anonymous Referee #2 · 23 Dec 2018

The manuscript titled "Characterising the Seasonal and Geographical Variability of Tropospheric Ozone, Stratospheric Influence and Recent Changes" presents a very interesting analysis on the stratospheric influence on tropospheric ozone using two chemistry-climate models CMAM and EMAC, as part of the IGAC/SPARC CCMI activity. The manuscript first shows that both models agree quite well with the observations from Satellite with the Ozone Monitoring Instrument (OMI) and from ozonesondes. Then the manuscript focuses on the models to study the variability of tropospheric ozone, stratospheric ozone and the stratospheric intrusions in order to assess how much stratospheric ozone impact tropospheric ozone. A statistically significant increase in tropospheric ozone is found across much of the world. The role of the stratosphere-troposphere exchange to such ozone changes ranges from 25-30% at the surface and 50-80% in the upper troposphere-lower stratosphere.

Although the manuscript is not so easy to read and follow, it is well structured; in particular, the summaries of each main section are very much appreciated.

I would suggest minor revisions, mainly clarifications, before the manuscript could be published.

General comments:
I found one general information missing about the models. It is the inferred stratospheric influx as mentioned in Young et al., 2013 (Table 2) for other CCMI models. Could the authors add this information?

Young, P. J., Archibald, A. T., Bowman, K. W., Lamarque, J. F., Naik, V., Stevenson, D. S. et al.: Pre-industrial to end 21st century projections of tropospheric ozone from the Atmospheric Chemistry and Climate Model Intercomparison Project (ACCMIP). Atmos. Chem. Phys., 13(4), 2063-2090, doi:10.5194/acp-13-2063-2013, 2013.

Specific comments:
Line 1 p. 2: Could the authors give the period of time on which the change in ozone was calculated: 4-6 ppbv (5-10%).

Line 24 p.2: "background ozone" is used here, when I think it refers to "baseline ozone". According to the Hemispheric Transport of Air Pollution 2010 Part A paper and Cooper et al. (2014), "Baseline concentrations refer to observations made at a site when it is not influenced by recent, locally emitted or produced anthropogenic pollution. The term global or hemispheric background concentration is a model construct that estimates the atmospheric concentration of a pollutant due to natural sources only."

Cooper, O. R., Parrish, D. D., Ziemke, J., Cupeiro, M., Galbally, I. E., Gilge, S., ... & Oltmans, S. J. (2014). Global distribution and trends of tropospheric ozone: An observation-based review.

HTAP, T., 2010. Hemispheric Transport of Air Pollution 2010 Part A: Ozone And Particulate Matter, Air Pollution Studies No. 17.

Line 12-14 p. 3: I am not sure to understand where "seasonal minimum' comes from. According to Tang et al. (2016), the STE ozone flux in the Northern Hemisphere shows a maximum in late spring and early summer as well. Could the authors clarify the sentence?

Line 9 p. 5: [Typo] In "24, 6, 48 and 24 h", "24" is written twice.

Line 26 p. 5: [Typo] Change "Langrangian" to "Lagrangian"

Line 1 p. 9: Could the authors add references about the intercomparison campaigns between 1970 and 1990, for example Beekman et al. (1994).
I would have the same comment for the "evidence that the ECC sondes have greater precision […]".

Beekmann, M., Ancellet, G., Megie, G., Smit, H., Kley, D., 1994b. Intercomparison campaign of vertical ozone profiles including electrochemical sondes of ecc and brewer-mast type and a ground based uv-differential absorption lidar. J. Atmos. Chem. 19, 259e288.

Line 18 p. 9: The authors wrote, "A seasonal maximum in tropospheric ozone is evident in each hemisphere during spring, which is more pronounced in the Northern Hemisphere and extended in many regions through summer". According to Figure 1a, the Northern Hemisphere shows a seasonal maximum in spring and summer. In spring the maximum is rather seen above 80N. How confident are you on the retrieval of tropospheric ozone above 80N? Wouldn't be rather a stratospheric signal?
Could the authors add this particular polar region (>80N) where the spring maximum is seen?

Line 21 p. 9: Use of the parenthesis: "northward (southward)". This is not really a good structure and the authors tend to overuse it through the manuscript. I would suggest writing it without the parenthesis. That would be better English and more fluent for the reader.

Figure 1 (p. 10): I would suggest to change the maximum limit of the colorbar. Tropospheric ozone (1000-450 hPa sub-column) barely reach 35 DU at a maximum. I would suggest to change 50 DU to 35 DU. The geographical variability of tropospheric ozone will then be easier to see. Would the authors know what is happening above South Africa for JJA and SON? There is ozone values around 30 DU on the cost and above the ocean around but rather 20 DU on the continent, as there would be a continent/ocean barrier. It does not seem real.

Line 23 p. 12: Could the authors explain more, maybe with an equation, how they link the "interannual variability" and the "seasonal aggregates of the computed relative standard deviation (RSD) of the monthly mean O3". It is not obvious. The interannual variability seems to be the variability year to year. Why would the authors study seasonal composites of RSD as a metric for the interannual variability?

Line 14 p. 17: Section 4. Could the authors explain more the difference between O3S and O3F? If there is any equation used, I would suggest adding them to the text. It is not so clear.

Line 18 p. 33: "RSD values of over 10%". How does 10% compare with other values? It is not a clear evidence for the reader that it shows an influence of ENSO and QBO.

Line 24 p. 33: "Taking this information into account". To which information do the authors refer?

Line 33 p. 33: The sentence started at this line and finishes line 2 p. 34, I think the sentence could be shortened.

Line 34 p.33: "(no larger than 20%)", I would suggest removing the parenthesis and writing "with biases no larger than 20%".

Line 13 p. 34: What does "high sensitivity to the tropopause" mean?

Figure S4 p. 6: [TYPO] in the caption change "CMAN" to "CMAM"

---

## Author Comment (AC1) · 6 Feb 2019

**Author Response to Anonymous Referee #1 Comments**

We thank the reviewer for her/his helpful comments. Referee comments are given in black and author comments/actions in red.

 "The paper by Williams et al. "Characterising the Seasonal and Geographical Variability of Tropospheric Ozone, Stratospheric Influence and Recent Changes" utilises satellite and ozonesonde observations and two chemistry-climate models to investigate the stratospheric influence on tropospheric ozone. The authors conclude that the influence of stratospheric on tropospheric ozone is larger than previously thought. The authors also assessed the tropospheric ozone over the periods of 1980-89 and 2001-2010, and find an overall significant increase in tropospheric ozone, and attribute 25-30% changes at the surface and 50-80% in the upper troposphere to the stratosphere-troposphere exchange. The paper is well written, and the analysis is generally thorough, but some clarifications and improvements are needed before the paper can be accepted for publication in ACP. Detailed comments are listed below."

Thank you for your positive assessment of our manuscript. We hope to have clarified the issues raised with details on changes given below.

 "General comments:

A major concern is that this study includes only two CCMI model results, which reduces the robustness of the finding, especially in that "the influence of STE in the tropospheric ozone is larger than previously thought". Furthermore, using only simulations constrained with prescribed dynamics might obscure the changes due to dynamical feedbacks, especially when assessing long-term changes in ozone. Therefore, I suggest that the authors tone down the conclusion mentioned above, and instead focus on the uncertainty in the contribution of STE to the tropospheric ozone. The limitation of using prescribed dynamics CCM simulations should also be noted and discussed. A wider usage of CCMI models would address the first comment. As a minimum, the authors should give a reason for using only the two chosen models."

We acknowledge your concern that there are caveats with confining the analyses to only two models (such as the robustness of findings) but we argue that such detailed analyses would not be possible in the context of a wider range of CCMI models, at least not within the scope of this paper alone. However the main reason for our choice of these two models is the availability of the $O_3S$ tracer which is defined similarly. Whilst some (but not all) of the other CCMs in CCMI have $O_3S$ tracer simulations, they are not defined equivalently. We also acknowledge your point regarding the use of the specified dynamics simulations and agree that the above conclusion needs to be toned down accordingly, with greater emphasis on the uncertainty in the contribution of STE to tropospheric ozone. Whilst we are fully aware that the use of specified dynamic simulations can supress chemistry-climate feedbacks, the constraint on dynamics is critical for our quantification of historical changes due to the primary influence of transport and dynamics on the chemical distribution of ozone VMR. We agree that the limitations of using only two CCMs and specified dynamics simulations needs to be noted and discussed more widely, which we now address in the revised manuscript. In response to the first comment, we now make clear the reason for our choice of the two models.

"Regarding previous studies, I doubt that the paper by Lamarque et al. (1999) is still a very relevant reference that the authors focus their comparisons on, given that the approach used in Lamarque et al. (1999) was very simplistic compared to what can be achieved using more recent state-of-the-art CCMs. Also, there are a few more studies that have investigated the impact of STE on tropospheric ozone which the authors failed to cite, for example, Lelieveld and Dentener (2000), Hess et al. (2013), etc. Jos Lelieveld and Frank J. Dentener, What controls tropospheric ozone? JGR, 105, P3531-3551 2000.

P. G. Hess and R. Zbinden, Stratospheric impact on tropospheric ozone variability and trends: 1990–2009, ACP, 13, 649–674, 2013.

Therefore, a more thorough review of the recent literature would be desirable."

Our explicit reference to Lamarque et al. (1999) relates to the similarity of their analyses to ours and we would argue that the comparison to their results highlights how our understanding of the influence of the stratosphere on tropospheric ozone has changed in the last twenty years as models have become increasingly more complex. However, we certainly agree with your point and thus also feel that other more recent publications on the matter (such as the ones suggested above) need mentioning, which will help for a more thorough review of the recent literature on the impact of STE on tropospheric ozone. This has now been implemented.

"Specific comments:

P2, L13: add "large" before "number""

Done.

"P4, L17: please add references here"

A reference has been added to support this point.

"P5, L5-L20: Can you describe the models' characteristics in a more objective way here? Why do you choose these two models specifically (there are quite a few other models from CCMI that you could include)? Also describe the main differences between these two models."

Please see paragraph for amendments to help describe such characteristics more objectively. A sentence has been added to justify the use of only the two models (based on the definition of the $O_3S$ tracer which is similar in both models and is either absent or defined different in other CCMI models. A sentence has also been added highlighting the main differences between the two models which is logically structured in more detail in the following specific model sub-sections (2.1.1 and 2.1.2).

"P5, L28: Pease provide more details in chemical schemes used in EMAC."

Please see following added sentence detailing the chemistry included in the MECCA chemistry submodel which goes into EMAC.

"P6, L4 & L21: Please provide more detailed information on how the O3S tracer is defined in terms of its chemical and dynamical nature in both models."

Please see manuscript for additional added information on the chemical and dynamical constraints on defining the O3S tracer.

"P7, L19-L21: Please clarify if the AKs have or have not been applied to the modelled and ozonesonde data when you compare these two. It only makes sense to apply AKs when compare model/sonde data to the satellite data."

No AKs have been applied for this comparison and rightly so as it only makes sense to apply AKs for satellite-model/satellite-sonde comparisons as you state. The sentence describes satellite-sonde validation efforts as reported by another study (Miles et al., 2015). Although this was cited further up, it has been cited again to avoid any confusion.

"P7, L27-L29: I don't understand why "The 1000-450 hPa (0-5.5 km) OMI subcolumn data is considered a representative approximation of the full tropospheric ozone column amount, due to vertical smearing of information from above 450 hPa (_ 5.5 km)." is the case. Is it possible to show AKs?"

We remove this claim as the accuracy of this representativeness is strongly latitude and seasonally dependent and due to lack of supporting literature to support this statement.

"P9, L1-L4: Please add references here. There are a series of publications on JOSIE by Smit et al.

Smit, H. G. J., and D. Kley (1998), JOSIE: The 1996 WMO International intercomparison of ozonesondes under quasi flight conditions in the environmental simulation chamber at Ju´llich, WMO Global Atmosphere Watch report series, No. 130 (Technical Document No. 926). World Meteorological Organization, Geneva.

Smit, H. G. J., and W. Straeter (2004a), JOSIE-1998, Performance of ECC Ozone Sondes of SPC-6A and ENSCI-Z Type, WMO Global Atmosphere Watch report series, No. 157 (Technical Document No. 1218), World Meteorological Organization, Geneva.

Smit, H. G. J., and W. Straeter (2004b), JOSIE-2000, Ju´llich Ozone Sonde Inter-comparison Experiment 2000, The 2000 WMO international intercomparison of operating procedures for ECC-ozonesondes at the environmental simulation facility at Ju´llich, WMO Global Atmosphere Watch report series, No. 158 (Technical Document No. 1225), World Meteorological Organization, Geneva."

Many thanks, these references have been added.

"P10, L19: Ozone is not at its minimum in SON in the SH, but maximum. Biomass burning emissions and STE usually dominate the seasonality of SH O3."

We acknowledge this is the case already for the SH as a whole, but this sentence refers only to Antarctica and the Southern Ocean where the influence of stratospheric ozone depletion is 'likely' dominant.

"P11, L21-23: can you provide more details on the difference between these two NOx emission datasets?"

Further details added courtesy of the Hoesly et al. (2018) reference.

"P12, L1-L2: This seems slightly mis-leading on the function of the AKs. The purpose of applying the AVKs is to compare like with like."

We use Fig.2 (Table S1) to show and describe the importance of applying AKs and the effect it has on tropospheric ozone which we would argue is informative for an uninitiated reader to AKs and the necessity of its application for like for like model-satellite measurement comparisons. Sentenced revised slightly to avoid misleading the reader.

"P12, L4-L7: Similar features seem exist in both models; it seems more likely due to transport barrier than STE (which the STE maximises in winter)."

We agree with this statement and revise this rather speculative sentence to allude to this as a possible cause, as well as the magnitude of the stratospheric ozone hole which could explain the retention of this feature after applying AKs.

"P12, L7-L9: Does the difference in chemical schemes between the two models play a role here?"

There could be differences due to disparities in the implementation of emissions in the model schemes and different treatments (e.g. bulking of species in CMAM) but we cannot disentangle such influence apart from dynamics here in our evaluations. We now acknowledge this as having a possible influence in the biases in each model here and allude to the need for further investigation.

"P13, Fig 3: It is impossible to discern the RSD of ozonesonde data denoted as circles, due to a uniformed colour scale."

We have revised the colobar scale down from 0-20 % to 0-14 % to make clearer structure in the RSD plots. Ozonesonde RSD is now made more easily distinguishable (white outline) but note that the RSD for sondes is uniformly low with few exceptions. The reason for this unclear and would warrant further investigation.

"P14, Fig 4: The value of 100 ppbv O3 seems a bit too low for defining the tropopause. Using 100 ppbv O3 also deviates from the definition by Bethan et al. (1996) (cited in caption), which is based on the ozone gradient, defined as the minimum altitude where the vertical gradient of the O3 VMR is greater than 60 ppb/km, remains so for a further 200 m, and the O3 mixing ratio is greater than 80 ppbv, exceeding 110 ppbv immediately above the tropopause."

Thank you for spotting this error, we now remove the citation to Bethan et al. (1996) and simply state that we choose the 100 ppbv ozone isopleth as a rough approximation of the chemical tropopause height.

"P15, L16-L32: Please note the figures that you are discussing throughout this paragraph.

Also, the description/discussion in this paragraph can be simplified to focus on the main points."

All discussion in this paragraph refers to Fig. 4 and we now remind the reader at the start of this paragraph. We have revised and shortened this paragraph for easier reading and to make clearer the main points.

"P16, L1-L2: again, please can you refer to the figure(s) that you are discussing."

Done.

"P16, L27: Do you also apply AVKs to model data when compare them with ozonesonde data? If so, it is not necessary."

No we do not. We merely use the model-ozonesonde comparison to infer how the biases arise/change between OMI and the models as a result of applying AKs to the models in these comparisons (Fig. 1 and 2).

"P16, L28: do you mean "simplified" tropospheric chemistry scheme? It is unusual to use the word "conservative"."

Yes, word changed.

"P16, L29-L31: which comparison/figures you are talking about here? Please make it clear by referring to figures."

The additional smearing (increase in subcolumn ozone due to AK application) can be seen in Fig. 2 but the conclusion is made based on the model-ozonesonde comparison (Fig. 4) as should be clear from the previous few sentences. We however remind the reader that the influence of the AK can be seen in Fig. 2 to make this clearer.

"P16, L32-L33: It seems lacking context regarding "since vertical smearing of information is far more limited due to a higher tropopause.". Could you be specific? Where is the information regarding a higher tropopause?"

We base our statement on the higher climatological mean position of the tropopause in the tropics/sub-tropics compared with the extratropics which will result in less smearing of information from the stratosphere. Sentenced revised for clarity.

"P17: L1: "must induce" should be "must have induced""

Changed.

"P17: L3-L7: Showing the AKs might help with the discussion here."

We add in an example of the OMI monthly mean AKs for August 2007 (~ 47°S) to illustrate this point in to the supplement (new Fig. S2), which importantly shows that the 1000-450 hPa

subcolumn is sensitive to influence from ~ 150-450 hPa pressure range. This indicates where the smearing of information can originate from. We refer the reader to this figure and emphasise this point in the manuscript.

"P21, L31: Please provide details on how you map the model data to ozonesonde measurements shown in Fig 7?"

Additional detail has been added. Ozonesonde profile measurements were aggregated for each month and for 10 degree latitude bands, which were then subsequently averaged over all 31 years (1980-2010) over all longitudes (zonal average). Similarly to Fig. 4, measurements were interpolated and averaged between ±20 hPa for each pressure level (350, 500 and 850 hPa).

"P23, L1: it is too general to say that "… are evident in the contemporary CCM simulation" while only two CCMs are used here."

Dropped the word "contemporary".

"P24, L11: What do you mean "even lower tropospheric ozone"?"

We refer to the lower troposphere but can see how such confusion may arise. The phrase "even lower" has been removed and we refer now to such influence extending down into the lower troposphere.

"P24, L20: What is the rationale for choosing these longitudes?"

Whilst the choice is a little arbitrary to illustrate that calculated changes are very much height-dependent, both the 30°W and 30°E transects intersect the small region of negative change over central Africa, which varies according to model and pressure level in terms of magnitude and location. The 30°W transect also intersects Greenland where there is significant model disparity at 350 hPa during MAM (Fig. 8a and 10a), although the cross-sectional view (Fig. 9a and 11a) shows such large differences to be confined only to the uppermost part of the domain near to the tropopause). We find this to be consistent with differences in tropopause height found by Hegglin et al. (2010) and our own finding that the lower branch of the BDC is stronger in CMAM. The 90°W transect (Fig. 9c and 11c) intersects the Himalayas where there are some interesting differences in the calculated changes on either side of this mountain range for both $O_3$ and $O_3S$ and between each model. Overall the selected transects help the discussion in this section to understand the upper level of stratospheric contribution to changes in near-surface ozone (largest across the Eurasian continent) and the source of such features such as the negative trend region over Africa (not evident in $O_3S$). A shorter version of this rationale is now added here in the manuscript for clarity.

"P25, Fig 8: There are large areas in the SH that are denoted significant in CMAM (SON 500 hPa and SON surface plots), which are not reflected in the relevant discussion. Please check."

We now acknowledge this in the manuscript but do not discuss in depth as changes are small. The modest increase would appear to be related to long range transport from the SH

subtropics and entrainment hemispherically by upper level winds especially. O₃S shows no such significant changes implying this increase is driven by changes in the troposphere.

"P27, L22: Please note which figure(s) you are discussing here? Is Fig 8?"

It is Fig. 8, yes. This has been added.

"P30, L10-L12: is "subtle shifts in the height of tropopause" shown anywhere?"

We do not show it anywhere but we do refer to Hegglin et al. (2010) earlier on in 4.1 (page 19, line 8) which finds that the CMAM tropopause is lower (some 30-50 hPa higher in pressure) than EMAC. This citation is now mentioned here also to support this statement.

"P34, L10-L12: What are the reference variables for these percentage changes?"

The percentage change values relate to the change values in ppbv we summarised earlier in 5.1 – e.g. The O₃ (Fig. 8) increase in the NH mid-latitudes during MAM/SON in the upper troposphere, being on the order of some 4-6 ppbv is seen to equate to a 1-3 ppbv increase in O₃S (Fig. 10) over the time period considered, hence we arrive at a ~ 25-50 % stratospheric contribution to the total change. We now mention here that such values are arrived at in conjunction with use of the tagged stratospheric (O₃S) tracer.

 "P35, L2: Please specify re "some regions of the world"."

We now give examples where the increase is substantial and generally significant for both models in one or more seasons: W. Eurasia, E. North America, S. Pacific Ocean and the S. Indian Ocean.

---

## Author Comment (AC2) · 6 Feb 2019

**Author Response to Anonymous Referee #2 Comments**

Thank you for your comments. Referee comments are given in black and author comments/actions in red.

"The manuscript titled "Characterising the Seasonal and Geographical Variability of Tropospheric Ozone, Stratospheric Influence and Recent Changes" presents a very interesting analysis on the stratospheric influence on tropospheric ozone using two chemistry-climate models CMAM and EMAC, as part of the IGAC/SPARC CCMI activity. The manuscript first shows that both models agree quite well with the observations from Satellite with the Ozone Monitoring Instrument (OMI) and from ozonesondes. Then the manuscript focuses on the models to study the variability of tropospheric ozone, stratospheric ozone and the stratospheric intrusions in order to assess how much stratospheric ozone impact tropospheric ozone. A statistically significant increase in tropospheric ozone is found across much of the world. The role of the stratosphere-troposphere exchange to such ozone changes ranges from 25-30% at the surface and 50-80% in the upper troposphere-lower stratosphere.

Although the manuscript is not so easy to read and follow, it is well structured; in particular, the summaries of each main section are very much appreciated.
I would suggest minor revisions, mainly clarifications, before the manuscript could be published."

Thanks for your feedback. Hopefully implementation of the suggested minor revisions/clarifications will help to improve the legibility of the manuscript.

"General comments:

I found one general information missing about the models. It is the inferred stratospheric influx as mentioned in Young et al., 2013 (Table 2) for other CCMI models. Could the authors add this information?

Young, P. J., Archibald, A. T., Bowman, K. W., Lamarque, J. F., Naik, V., Stevenson, D. S. et al.: Pre-industrial to end 21st century projections of tropospheric ozone from the Atmospheric Chemistry and Climate Model Intercomparison Project (ACCMIP). Atmos. Chem. Phys., 13(4), 2063-2090, doi:10.5194/acp-13-2063-2013, 2013".

This information is only from ACCMIP (a subset of CCMI models) and therefore we summarise such information instead from table 8.1 from the IPCC WG1 AR5 report which includes the mean stratospheric influx from this subset of models, in addition to a range of other models and observational estimates. Sentence added in opening paragraph of section 2.

"Specific comments:

Line 1 p. 2: Could the authors give the period of time on which the change in ozone was calculated: 4-6 ppbv (5-10%)".

This information should have been implicit as the time period was mentioned in the previous sentence (line 34-35, p. 1). Have omitted this detail here and given the period of time the change was calculated over at the end of the above sentence (line 2, p. 2) for clarity.

"Line 24 p.2: "background ozone" is used here, when I think it refers to "baseline ozone". According to the Hemispheric Transport of Air Pollution 2010 Part A paper and Cooper et al.

(2014), "Baseline concentrations refer to observations made at a site when it is not influenced by recent, locally emitted or produced anthropogenic pollution. The term global or hemispheric background concentration is a model construct that estimates the atmospheric concentration of a pollutant due to natural sources only.

Cooper, O. R., Parrish, D. D., Ziemke, J., Cupeiro, M., Galbally, I. E., Gilge, S., ... & Oltmans, S. J. (2014). Global distribution and trends of tropospheric ozone: An observation-based review.

HTAP, T., 2010. Hemispheric Transport of Air Pollution 2010 Part A: Ozone And Particulate Matter, Air Pollution Studies No. 17".

Thanks for pointing this out. We indeed misuse the term 'background ozone' as referring to the 'baseline ozone' when citing Cooper et al. (2014) and so have corrected this and added in the additional HTAP (2010) reference. Our reference to studies which refer to 'background ozone' remain, but we keep these citations separate from the two above.

"Line 12-14 p. 3: I am not sure to understand where "seasonal minimum' comes from. According to Tang et al. (2016), the STE ozone flux in the Northern Hemisphere shows a maximum in late spring and early summer as well. Could the authors clarify the sentence?"

The phrase "seasonal minimum" on line 13 relates to the STE mass flux, not the STE ozone flux which we acknowledge has a seasonal maximum in late spring and early summer (in agreement with Yang et al. (2016)?) on line 12. Have revised this sentence, which hopefully clarifies this better and improves readability.

"Line 9 p. 5: [Typo] In "24, 6, 48 and 24 h", "24" is written twice".

This is an actual fact not a typo. These times refer to the nudging to temperature, vorticity, divergence, and surface pressure respectively. These are now listed following a colon and taken out of parentheses to avoid confusion.

"Line 26 p. 5: [Typo] Change "Langrangian" to "Lagrangian""

Typo corrected.

"Line 1 p. 9: Could the authors add references about the intercomparison campaigns between 1970 and 1990, for example Beekman et al. (1994). I would have the same comment for the "evidence that the ECC sondes have greater precision […]".

Beekmann, M., Ancellet, G., Megie, G., Smit, H., Kley, D., 1994b. Intercomparison campaign of vertical ozone profiles including electrochemical sondes of ecc and brewer-mast type and a ground based uv-differential absorption lidar. J. Atmos. Chem. 19, 259e288."

References added, many thanks.

"Line 18 p. 9: The authors wrote, "A seasonal maximum in tropospheric ozone is evident in each hemisphere during spring, which is more pronounced in the Northern Hemisphere and extended in many regions through summer". According to Figure 1a, the Northern Hemisphere shows a seasonal maximum in spring and summer. In spring the maximum is rather seen

above 80N. How confident are you on the retrieval of tropospheric ozone above 80N? Wouldn't be rather a stratospheric signal?
Could the authors add this particular polar region (>80N) where the spring maximum is seen?"

We make no change to the manuscript as the retrieval should not be trusted at these latitudes due to the influence of the OMI row-anomaly (rows on the 2-D detector which have become damaged or blocked by insulation blankets – mentioned on page 8, L6-9). We have instead extended the grey masking to cover this region in MAM/JJA.

"Line 21 p. 9: Use of the parenthesis: "northward (southward)". This is not really a good structure and the authors tend to overuse it through the manuscript. I would suggest writing it without the parenthesis. That would be better English and more fluent for the reader.

Sentence revised and we limit use of this structure elsewhere to only sentences where we deem appropriate and fluency is not compromised for the reader.

Figure 1 (p. 10): I would suggest to change the maximum limit of the colorbar. Tropospheric ozone (1000-450 hPa sub-column) barely reach 35 DU at a maximum. I would suggest to change 50 DU to 35 DU. The geographical variability of tropospheric ozone will then be easier to see.
Would the authors know what is happening above South Africa for JJA and SON? There is ozone values around 30 DU on the cost and above the ocean around but rather 20 DU on the continent, as there would be a continent/ocean barrier. It does not seem real."

We have revised this colour scale accordingly. We believe advection of precursor-rich air from the continent (due to biomass burning) and later formation of photochemically formed ozone to explain the higher values offshore, together with the reduced depth of the subcolumn over the relatively elevated South African mainland.

"Line 23 p. 12: Could the authors explain more, maybe with an equation, how they link the "interannual variability" and the "seasonal aggregates of the computed relative standard deviation (RSD) of the monthly mean O3". It is not obvious. The interannual variability seems to be the variability year to year. Why would the authors study seasonal composites of RSD as a metric for the interannual variability?"

We have added an equation immediately below this sentence for clarity in how we calculate seasonal composites of RSD. The standard deviation is normalised with respect to the mean of each individual month over all years (2005-2010) to compute the monthly RSD which we then aggregate by calendar season. This metric therefore captures variability with respect to the monthly resolved seasonality over all years which we infer here as the interannual variability.

"Line 14 p. 17: Section 4. Could the authors explain more the difference between O3S and O3F? If there is any equation used, I would suggest adding them to the text. It is not so clear."

We now make this clear through revising this sentence and have also added the $O_3F$ equation in line (in addition to the Fig. 5 caption).

"Line 18 p. 33: "RSD values of over 10%". How does 10% compare with other values? It is not a clear evidence for the reader that it shows an influence of ENSO and QBO."

According to Lee et al. (2010), a study that was originally cited in section 3.2, the QBO is estimated to induce anomalies of as much as 10-20 % in tropical tropospheric ozone, which would scale with the 10-20 % variability in RSD found in both OMI and the models. This is again referred to here to ensure the reader makes this connection but 'ENSO' is dropped as the study found that the influence from ENSO is likely much smaller (~ 3 %).

"Line 24 p. 33: "Taking this information into account". To which information do the authors refer?"

This refers to the findings of section 3 (summarised in the paragraph above in section 6: conclusions). This is made clearer to the reader.

"Line 33 p. 33: The sentence started at this line and finishes line 2 p. 34, I think the sentence could be shortened."

This sentence has been split in two.

"Line 34 p.33: "(no larger than 20%)", I would suggest removing the parenthesis and writing "with biases no larger than 20%"."

This has been revised.

"Line 13 p. 34: What does "high sensitivity to the tropopause" mean?"

This refers to the height of the tropopause being key to the calculated changes (due to the associated sharp vertical gradients in ozone). Changed to 'known discrepancies in tropopause height' as found in Hegglin et al. (2010).

"Figure S4 p. 6: [TYPO] in the caption change "CMAN" to "CMAM""

Thanks, typo corrected.

---

## Author Comment (AC3) · 6 Feb 2019

**Author Response to Anonymous Referee #3 Comments**

Thank you for your comments. Referee comments in black and author comments/actions in red.

"This is an interesting and useful analysis, but needs to be put in context and contrasted with other recent work before it is published. The authors would also be well-advised to better qualify the limitations of their study, in particular with regard to tropospheric chemistry."

Thanks, we agree with these comments and now have added more references to contemporary work and better pointed out the limitations of our study, particularly relating to tropospheric chemistry and the specified dynamics simulations used.

"Detailed remarks:

Page 3, lines 21-25: (and elsewhere) the authors focus on Lamarque et al. (1999), a 20-year old study. There are much more recent modeling studies that show larger net influence of STE. For example, Figure 6 of Banerjee et al. (2016) looks very much like Figure 5 of this manuscript. Doubtless similar plots for other models exist. If the authors wish to make the case for their result "that the stratospheric influence on tropospheric ozone is larger than previously thought", they need to quote recent studies that show smaller influence."

We agree. The statement has been toned down to reflect the important role of the stratosphere more generally. The influence of STE on tropospheric ozone is now updated here in accordance with more recent studies (including a reference to Banerjee et al., 2016). However, we keep the reference to Lamarque et al. (1999) in places to emphasise the similarity of their analyses to ours and the focus of the paper.

"Line 34: Similarly, the authors cite only one observational study, and for the Southern Ocean. There have been many such studies, e.g. Dibb et al. (1994); Elbern et al. (1997); Stohl et al. (2000); Zanis et al. (2003); Colette and Ancellet (2005); Cooper et al. (2006); Thompson et al. (2007a,b); Cristofanelli et al., (2010); Tarasick et al., (2018). Citing some of these would not only be appropriate, but would support the authors' point, as in general they find modest influence of STE on lower tropospheric ozone levels."

Thanks for these additional references. We now summarise and discuss this literature in a separate paragraph which immediately follows on from the discussion of stratospheric influence according to model studies.

"Page 4, lines 10-11: While I agree that there are certainly major limitations with the accuracy of retrieved tropospheric ozone from spaceborne instruments, I take issue with the statement "…scientists must instead rely on tools such as chemistry climate models (CCMs) to fill in the gaps in our understanding of the global distribution of tropospheric ozone". NO, NO, NO! Models are sophisticated data visualization tools: they contain (at best) all that we know about the atmosphere. They allow us insights and interpretation that would not be possible without them, but they do not contain anything that we don't, collectively, already know."

Whilst we largely agree with this point, we would argue that quantification of the stratospheric contribution ($O_3S$) to both the vertical and global distribution of tropospheric ozone is not

possible from observations alone. Additionally, specific model simulations and diagnostics can help to disentangle various feedbacks/mechanisms that could not be inferred from observations alone. We have rewritten this statement to now better explain the added scientific value we can gain from CCMs in comparison with observations.

"Page 8, lines 10-11: I thought the main problem was lack of photons that actually penetrate this far, as well as lack of contrast in the scattered spectrum, compared to a few km higher up."

We agree that this is the main issue with the retrieval of particularly lower tropospheric ozone (with errors in the retrieval due to albedo and aerosols for instance of secondary importance). This detail has been added into the manuscript.

"Page 8, line 18: It seems odd to cite satellite papers for generic facts about the global ozonesonde network. Liu et al. (2013b) has a good discussion, with a map and table of sites. In line 26, the proper reference here is Smit et al. (2007), although the others are fine as additional references. On the next page (line 2), citations are required for the WMO & JOSIE campaigns."

We agree with this point and omit some of this detail, referring the reader to the suggested Liu et al. (2013b) reference. Citations have been added for the WMO and JOSIE campaigns. Many thanks for these references and clarification.

"Page 9: I find the statement in lines 28-30 quite remarkable. I believe there are many studies showing that long-range transport of ozone and its precursors are the dominant source of ozone in remote areas."

We agree that this argument is valid but more applicable to the lower troposphere. We thus revise this sentence to refer to the free troposphere and tone down our assertion of the dominant role of STE to avoid any contradiction.

"Page 17 (top paragraph), and elsewhere: the authors put a lot of effort into explaining the effects of "vertical smearing". Of course they need to consider OMI AKs when comparing to OMI data, but perhaps they would find it easier to use a 3D ozonesonde based dataset, like Liu et al. (2013a,b)."

We believe our paper will serve to highlight the importance of AKs for model-satellite measurement comparisons which is sometimes not fully understood or appreciated within the CCMI community. We agree that comparisons with an ozonesonde-based dataset using trajectory mapping would provide further insight but of course such products have their limitations. Such analysis we would suggest is beyond the scope of this study but we now mention such approach could be warranted to further establish and confirm the presence of such model biases we found in our model-ozonesonde comparison (Fig. 4).

"Page 22 (Figure 7a): How are these plots produced from ozonesonde data?"

This is now made clear in the opening sentence to sub-section 4.3. Ozonesonde profile measurements were aggregated for each month and for 10 degree latitude bands, which were

then subsequently averaged over all 31 years (1980-2010) over all longitudes (zonal average). Similarly to Fig. 4, measurements were interpolated and averaged between ±20 hPa for each pressure level (350, 500 and 850 hPa).

"Page 23, line 22: The "significant difference in the strength and dominance of the shallow branch of the BDC in each model" needs more explanation. It is first mentioned here, in the Summary."

The build-up and burden of ozone in the extratropical lowermost stratosphere is directly related to strength of the lower BDC branch (since the equatorial region is where most ozone is produced). We add in this additional detail and include citations.

"Page 31 (Summary): The authors should consider, and discuss, the differences between their results and the much smaller STE response found by Neu et al. (2014). In particular, Neu et al. claim that larger responses of tropospheric ozone to STE are found in models without comprehensive tropospheric chemistry."

We have added a few sentences discussing our findings in relation to the earlier study by Neu et al. (2014) and note this important caveat.

"Page 32, lines 31-34: Discussing the comparison before the AKs are applied makes little sense, and should be omitted."

Actually, we feel this finding highlights an important trade-off in applying AKs to models that have known stratospheric biases for model-measurement comparisons of tropospheric ozone. In the case of CMAM, we can infer through our analyses (Fig.2 and Fig. 4 in section 3) that the closer agreement to OMI arises due to the competing influence of the relatively simple tropospheric chemistry scheme (underproduction of in situ photchemical formation of ozone) and excessive smearing of stratospheric ozone due to a high bias in the lower stratosphere (~ +20-60 %). Such analyses therefore show that CMAM is more deficient in its representation of tropospheric ozone than EMAC, whereas the opposite might be inferred from Fig. 1 alone. We expand this point to argue our case and make the reader aware that in a limit number of cases, where stratospheric biases are sufficiently large, application of AKs for model-measurement comparisons of tropospheric ozone would not be advocated, particularly if the model representation of tropospheric ozone is known to be good.

"Page 33, line 26: See previous comment, page 23, line 22."

Sentence expanded to explain this conclusion again.

"Minor points:

Page 5, lines 11, 12: Typographical errors."

Sentence has since been removed.

"Page 5, line 34: The solar cycle evolves?"

We refer to the 11-year solar cycle which we now state explicitly for clarity.

"Page 6, line 9: Typographical error."

Removed 'have'.

"Page 6, line 19: "Compared with EMAC"?"

Changed to 'In contrast to EMAC'

"Page 8, line 20: Actually the WOUDC also has data for Indian, Brewer-GDR, carboniodine and Regener sondes."

Thanks for this clarification. We however remove such detail and direct the reader elsewhere (e.g. Liu et al., 2013b) as suggested by yourself.

---

## Author Comment (AC4) · 6 Feb 2019

**Author Additional Comments**

Referee comments in black and author comments/actions in red.

Further amendments to the manuscript have been made in accordance with an internal review procedure within the Canadian Centre for Climate Modelling and Analysis department of Environment Canada, of which is the host institution of co-author *David A. Plummer*. These amendments are listed below:

"P1, L32: With the model biased high in the lower strat and underproducing photochemical O3 in the troposphere, the conclusion that stratospheric O3 intrusions are larger than previously thought is cast in doubt, since that may just be because of the model biases.

Can more explanation be added here to explain why the authors still feel confident in that conclusion? For example, was a correction applied to the model before coming to that conclusion?"

Changed "larger than previously thought" to "significant". Although we can be confident that the real influence of the stratosphere is larger than that found by Lamarque et al. (1999), which is the main study we base this assertion on, we have to appreciate it is a twenty year old study that was based on a much simpler model. We therefore include new references throughout the manuscript as suggested by the reviewers and tone down statements such as that above. We see no reason to alter the estimated influence exceeding 50 % in the wintertime high latitudes however, as both model show this (it was only CMAM which had the high (low) bias in the lower stratosphere (troposphere) and if anything the inverse was found for EMAC). However, it is an important point and does suggest there is some uncertainty still in our understanding of the stratospheric influence.

"P2, L25: Fiore et al (2002, JGR) and references therein could be added as older (more seminal) references of the increase in background tropospheric O3.

Fiore, A. M., D. J. Jacob, I. Bey, R. M. Yantosca, B. D. Field, A. C. Fusco, and J. G. Wilkinson (2002), Background ozone over the United
States in summer: Origin, trend, and contribution to pollution episodes, J. Geophys. Res., 107(D15), ACH 11–1–ACH 11-25, doi:10.1029/2001JD000982."

Done.

Here's another reference for transpacific transport of O3:

"P2, L28-29: Zhang, L., et al. (2008), Transpacific transport of ozone pollution and the effect of recent Asian emission increases on air quality in North America: An integrated analysis using satellite, aircraft, ozonesonde, and surface observations, Atmos. Chem. Phys."

Reference added.

"P4, L7: Lines 5-11 read as a little dismissive of the many excellent tropospheric O3 measurements from satellites such as TES, OMI/Trop-OMI, TOMS, MLS, etc - many of which have \*long\* term (> decade), global datasets, and are well validated by ground-based remote sensing, in situ, and inter-satellite comparisons.

If just trying to motivate the use of models to fill in the gaps in measurements, one may mention that the satellite measurements are usually limited to just a couple of overpass times per day at each location, and often have large errors in retrieved O3 (but are those uncertainties more than those from the mode??).
However, the second point about assessing and quantifying the causes and processes of trop O3 is good".

These lines have been rewritten to sound less dismissive, with greater acknowledgement of the value such satellite datasets provide and their contribution to our understanding of tropospheric ozone. The paragraph as a whole has also been modified to highlight the value of CCMs in terms of providing insights and interpretation (mechanisms, feedbacks and quantification of the stratospheric influence) that cannot be inferred from observations alone, as opposed to the filling in of gaps in the measurements, which is not such a valid point for such a reason you mention.

"P4, L7-8: add references"

Reference added.

"P4, L16-17: do you have a reference or other proof (e.g., your own calculations) of emissions being the largest source of model uncertainty?"

No, but we add in a reference which supports this claim.

"P4, L27: what does F stand for? Is O3F the total O3? ...Ok, coming back to this, I think I understand what O3F is, and "F" may stand for "fraction", perhaps?

Ideally O3S and O3F would be changed to something clearer like: O3_strat and O3_%strat (use subscript instead of "_")

The F indeed refers to fraction. This has been made clearer in the manuscript and indeed should have been defined as such here (the first instance it is mentioned).

"P6, L9: EMAC also extends to 0.01 hPa, but above you haven't given the altitude. Either move to first mention of 0.01 hPa, or remove the km altitude altogether, since that's highly variable depending on latitude".

Removed 95 km reference.

"P13, Fig. 3: The RSD from the ozonesondes looks to be much lower than that from OMI (and that from the models too), but this is not discussed in Section 3.2."

The scale range has been revised to show more structure in the RSD plots, however the ozoneosnde RSD is significantly lower than OMI in particular with few exceptions. This will require further investigation.

"P14, L25-29: I think the ppbv goes with Fig S1, but this method of using brackets is confusing b/c there is also left, middle, and right in brackets in the same sentence. I suggest describing Fig 4 in one sentence, and then adding a second sentence, saying that Fig S1 shows the same thing, but absolute differences of VMRs instead of percent differences."

Correct but agreed that the use of brackets is excessive in this sentence so have revised this and split into two sentences in accordance with your suggestion. Many thanks.

"P15, L31: change "should be" to "are""

Done.

"P17, L23: It's not clear to me what O3S is. From the wording, it sounds like it is just O3. But from the equation given in the caption, O3S seems to be the tagged stratospheric O3. If the latter, this sentence needs to be changed to say "Seasonal composites of the monthly mean, zonal-mean vertical distribution of stratosphere-originating ozone (O3S)" or "...tagged-stratospheric ozone (O3S)".

This comes back to my earlier comment that O3F and O3S are not described clearly enough, and was confusing to me (and possibly other readers). Ideally, "O3F" and "O3S" would be replaced by
"O3_%strat" and "O3_strat""

This has been clarified and sentence broken up to avoid excessive use of brackets and enhance readability.

"P17, L24: How is Fig. S2 different from Fig 5? I don't see any corresponding bracketed statement that goes with S2 rather then Fig 5...
Please add the explanation in these brackets, and make it clearer in the caption for Fig S2. For example, the caption to Fig S2 could be "Figure S2 - same as Fig 5, but for ...[whatever is diffferent]."

...Ok, it took me a while, but now I see that the above sentence referring to O3 corresponds to Fig 5, and the bracketed "(O3S concentration)" corresponds to Fig S2. This method of using brackets to try to save wording makes it unnecessary confusing for the reader, and could be written just as efficiently as follows:

"Seasonal composites of [....] ozone concentrations (is it concentrations or VMR? clearly state one without using brackets) from 1000-80 hPa are shown in Figure 5 for EMAC (a), .... together with ... . The same is shown for the stratospheric-tagged O3 (O3S) in Figure S2.""

This has been remedied following the action taken immediately above.

P17, L33: There are alot of these bracketed inverse statements that I think should just be restated for clarity. For example, here it doesn't cost too many extra words to say:

"with the former clearly a greater influence near the surface, and the latter in the upper troposphere."

I would reword most, if not all, cases like this in the paper to improve clarity when it can be done so efficiently."

Done. We appreciate your point and make such revisions where necessary elsewhere to enhance readability and avoid reader confusion.

"P19, L25: Although this is a straightforward sentence using the bracket method, rewording to "when O3_%strat reaches a maximum in winter and minimum in summer." only adds one word ("and") and is clearer to the reader."

Done.

"P20, Fig. 6: what is the gray? Please mention what it is in the caption."

The grey shaded regions represent where surface pressure values are lower than the plotted pressure level (i.e. where each pressure level would be below ground). This is now indicated in the figure caption.

"P23, L1: "in the present study""

Phrase added.

"P24, L11: what is meant by "even lower"? Meaning the models are biased even lower? Or meaning even lower in altitude? This should be clarified b/c if interpreted the first way, then a low-biased tropospheric O3 will given erroneously high biased O3F in the troposphere.

I would remove "even lower" from the sentence."

We were referring to the influence exceeding 50 % as far down as the lower troposphere and hence we revise the sentence to make this clearer and avoid reader misinterpretation.

"P24, L19: just say "modelled O3 VMRs", remove the word "concentration" b/c VMR is not a concentration."

Done.

"P24, L19: again I'm confused: does Fig S5 correspond with O3 VMR and Fig 8 correspond to O3 concentration? The brackets are confusing and unnecessary."

Fig. S5 refers to the seasons DJF/JJA (also brackets at end of sentence). We agree that this method could confuse the reader so have clarified this.

"P24, L21: re-write to clearly state that Fig S6 is for DJF/JJA - if that's the case."

Again, this has now been clarified.

"P24, L22: (and Fig. S7 for ...)"

Again meaning for DJF/JJA. This is now clearer.

"P24, L23: ditto"

Also for DJF/JJA but for the cross-sections. This is now made more obvious for the reader.

"P24, L29: by what measure? ...The paired t-test p-value threshold should be interpreted with caution, and I suggest the authors add a caveat (e.g., reference to Waserstein & Lazar paper that I mention in a different comment)."

We now reference this citation and make the reader aware that such stippling needs to be interpreted with caution.

"P25, Fig. 8: A word of warning about interpreting the t-test threshold this way. The American Statistical Association (ASA) has thrown out the idea of using p-value thresholds to confirm or deny the null hypothesis, saying that you have to look at the broader picture and additional data to determine significance. Please see: "The ASA's Statement on p-Values: Context, Process, and Purpose", by Wasserstein, R.L., and Lazar, N.A., The American Statistician, 70:2, 129-133, 2016. https://www.tandfonline.com/doi/full/10.1080/00031305.2016.1154108 ...and consider revising wording in this paper to be less definitive about statistical significance based on the $p < 0.05$ threshold."

Thank you for bringing this to my attention. We add in the necessary caveats when discussing the stippled regions in relation to statistical significance.

"P30, L9: explain what's in Fig S8 in the brackets"

Specified the two different seasons presented in each of Fig. 11 (MAM/SON) and Fig. S8 (DJF/SON) for clarity.

"P31, L31: ....4-6 ppbv over the Northern Hemisphere and 2-6 ppbv over the Southern Hemisphere subtropics..."

Revised for greater clarity.

"P32, L3: too many brackets! Reword to remove as many as possible."

Most of these brackets have now been removed.

"P33, L6: why would a complex chemistry scheme cause a high bias? Do you mean "inaccuracies in the complex chemistry scheme"?"

You are right, it should not unless inaccuracies exist but I would not know if this is the case. The implication here is that the emission inventories will have an error attached and this might manifest more prominently in EMAC's modelling of ozone due to the complexity of the chemistry scheme. This would favour a high bias in this model if such inventories are an overestimate of the truth as implied by Hoesly et al., 2018. Sentence has been revised to reflect this.

"P33, L6: but both models used the same emissions, no?"
Correct, this has now been stated as so.

"P33, L16: Is there a paper on OMI's long-term performance that you can reference?"

We have added a reference that discusses the long-term performance of OMI – Levelt et al. (2018).

"P34, L4: due to ... ?"

Further detail from the Bonisch et al. (2011) reference has been added regarding the cause of reduced transit times in the lower stratosphere.